

# Dominant controls of transpiration along a hillslope transect inferred from ecohydrological measurements and thermodynamic limits

M. Renner[1], S. K. Hassler[2,6], T. Blume[2], M. Weiler[3], A. Hildebrandt[4,1], M. Guderle[4,1,7], S. J. Schymanski[5], and A. Kleidon[1]

[1]Max-Planck-Institut für Biogeochemie, Jena, Germany
[2]GFZ German Research Centre for Geosciences, Section Hydrology, Potsdam, Germany
[3]Universität Freiburg, Hydrologie, Freiburg, Germany
[4]Universität Jena, Ecological Modelling Group, Jena, Germany
[5]ETH Zürich, Soil and Terrestrial Environmental Physics, Switzerland
[6]Karlsruhe Institute of Technology, Institute of Water and River Basin Management, Karlsruhe, Germany
[7]Technische Universität München, Chair for Terrestrial Ecology, Department of Ecology and Ecosystemmanagement

*Correspondence to:* M. Renner (mrenner@bgc-jena.mpg.de)

**Abstract.**

We combine ecohydrological observations of sapflow and soil moisture with thermodynamically constrained estimates of atmospheric evaporative demand to infer the dominant controls of forest transpiration in complex terrain. We hypothesize that daily variations in transpiration are dominated by variations in atmospheric demand, while site-specific controls, including

limiting soil moisture, act on longer time scales.

We test these hypotheses with data of a measurement setup consisting of 5 sites along a valley cross-section in Luxembourg. Both hillslopes are covered by forest dominated by European beech (Fagus sylvatica L.). Two independent measurements are used to estimate stand transpiration: (i) sapflow and (ii) diurnal variations in soil moisture, which were used to estimate the daily root water uptake. Atmospheric evaporative demand is estimated through thermodynamically-constrained evaporation

which only requires absorbed solar radiation and temperature as input data without any empirical parameters.

Both transpiration estimates are strongly correlated to atmospheric demand at the daily timescale. We find that neither vapor pressure deficit nor wind speed add to the explained variance, supporting the idea that they are dependent variables on land-atmosphere exchange and the surface energy budget.

Estimated stand transpiration was rather similar between the north- and the south-facing hillslopes despite the different

aspect and the largely different stand composition. We identified an inverse relationship between sap flux density and the site-average sapwood area per tree as estimated by the site forest inventories. This suggests that tree hydraulic adaptation can compensate for heterogeneous conditions. However, during dry summer periods differences in topographic factors and stand structure can cause spatially variable transpiration rates.





## 1 Introduction

Evapotranspiration $E$ couples water and energy balances at the land surface and is constrained by both, the supply of water and the atmospheric demand for water. Total $E$ is composed of evaporation from intercepted water on plants and the surface, soil evaporation and the physiological process of plant transpiration ($E_T$) taking water from the soil (Shuttleworth, 1993).

Transpiration is of key importance for the (local) climate by altering the surface energy balance, and for water resources where $E_T$ is an important loss term of the water balance. The tight coupling to photosynthesis and thus primary productivity makes $E_T$ also central for agriculture and forestry. There is thus a need to understand the spatial and temporal variation of $E_T$ as a result of interacting biogeophysical processes. Especially in temperate forests these biogeophysical feedbacks are poorly understood, which strongly reduces our ability to assess their role in mitigating climate change (Bonan, 2008).

The agricultural need for fertile, flat terrain has led to a land-use pattern in Western Europe where forests are often found in complex topographical settings. Thus we have to consider that the first-order controls are strongly altered by topography: Hillslope angle and aspect systematically alter the amount of absorbed solar radiation and thus atmospheric demand (Baumgartner, 1960; Lee and Baumgartner, 1966), whereas slope and the topographic setting such as contributing area alter lateral redistribution of water (Famiglietti et al., 1998; Bachmair and Weiler, 2011). Apparently, forests adjust to these conditions,

but little is known how vegetation and site-scale transpiration in particular respond to these topographically altered boundary conditions of supply and demand (Tromp-van Meerveld and McDonnell, 2006). For example Holst et al. (2010) found that detailed forest hydrological models yielded different trends of simulated mean stand transpiration when comparing European beech stands on different hillslope aspects. These differences add up when simulating runoff generation from these sites and emphasizes the difficulty in the assessment of forest water balance in complex terrain.

Although detailed models of various feedbacks between plant physiology and the environmental conditions are available (Monteith, 1965; Farquhar et al., 1980; Wang and Jarvis, 1990; Whitehead, 1998; Haas et al., 2013) their applicability is generally restricted by the need for detailed physiologic, atmospheric and soil parameters. As an alternative to improve our understanding by increasingly detailed modeling, several authors proposed to deduce and predict $E$ through physical and physiological constraints (Calder, 1998; West et al., 1999; Raupach, 2001; Zhang et al., 2008; Wang and Bras, 2011). Applications

of these fundamental constraints do not only increase our understanding of the soil-plant atmosphere continuum, they may also lead to a few but independent predictors of $E$. For example, Kleidon and Renner (2013b) applied thermodynamic limits of convective heat exchange to the surface-atmosphere system. They found that under the assumption of maximum convective power and no surface water limitation, atmospheric demand is only dependent on absorbed solar radiation and surface temperature, being consistent with empirical formulations of potential evaporation of Makkink (Makkink, 1957) and the well-known

equilibrium evaporation concept (Schmidt, 1915; Priestley and Taylor, 1972). Contrary to the classic notion of Dalton evaporation, where vapor pressure deficit (VPD) and wind speed are used as forcing variables of potential evapotranspiration, Kleidon and Renner (2013b) argued that VPD and wind speed are rather dependent variables which emerge from the land-atmosphere interaction. The practical implication of maximum convective power is that atmospheric demand can be estimated without these variables and empirical parameters.





The approach of maximum convective power has so far only been applied for long-term annual means and large-scale geographic variations (Kleidon et al., 2014). Here we test this approach at the site level and for daily time scales in complex topographical terrain. We use data of a well instrumented beech forest with measurement sites across a transect with a north- and a south-facing hillslope. The indirect assessment of $E_T$ is based on ecohydrological measurements of sap flow and soil moisture. We present a systematic analysis to address the question of how much transpiration varies in complex terrain and what measurable site characteristics influence forest transpiration. In particular we hypothesize that potential evaporation representing the atmospheric demand for water can be estimated by thermodynamic limits of convection which only relies on surface absorption of solar radiation and temperature. Specifically, we test how much of the daily variations of in-situ transpiration observations can be explained by atmospheric demand and how the response to atmospheric demand changes along the measurement transect. Therefore we perform a linear regression and correlation analysis of the transpiration estimates to atmospheric demand comprising daily data of the growing season. Further, we evaluate how much additional variance can be explained by other time-varying parameters such as VPD, wind speed and soil moisture. Differences in the strength of correlation between sites would imply that site-dependent, time-varying constraints on transpiration are relevant. In contrast, a high, consistent correlation would imply that atmospheric demand and thus energy limitation of transpiration is the dominant driver of day to day variability. The overall response of transpiration to atmospheric demand represented by the slope of the linear regression would then indicate the importance of site dependent controls such as topographic, soil, and plant factors.

## 2 Methods

### 2.1 Site description

We analyze measurements at 6 different sites along a well instrumented steep forested hillslope transect (north- vs. south-facing, see Fig. 1) in the Attert catchment in Luxembourg over the vegetation period of 2013. The hillslope transect is part of the CAOS field observatory (Zehe et al., 2014) and is located in the western part of Luxembourg (5E°48'13", 49N°49'34") at about 460 m NN. The land cover of the transect is a mixed forest dominated by European beech (Fagus sylvatica L.). The north-facing slope has an inclination of $\approx 15°$ and is composed of a few dominant trees with large gaps and dense understorey mainly of young beech trees, whereas the south-facing slope is generally steeper ($\approx 22°$) and has no understorey and a denser canopy. Also tree species composition varies between slopes, with 97% beech on the north-facing sites with single spruce trees and 90% beech and 10% oak on the south-facing sites. The valley site has 80% beech with 10% spruce and alder, respectively. Geologically, the site is situated in northeast-southwest-trending fold system of Schists of the Ardennes Massif. The shallow soils developed on periglacial slope deposits (Juilleret et al., 2011). The deposits are generally found at a depth of 70-90 cm.

Standard meteorological data, global radiation, air temperature, relative humidity and wind speed were measured 2 m above ground at all sites. For the meteorological forcing we used the data from the open grassland site G1 which is located 240 m to the northwest of the forest site N1, see Fig. 1. Absorbed solar radiation $R_{sn} = (1 - \alpha)R_g$ in $W m^{-2}$ is derived from global radiation $R_g$ and an albedo estimate of $\alpha = 0.15$ which is representative for deciduous forests (Oke, 1987).



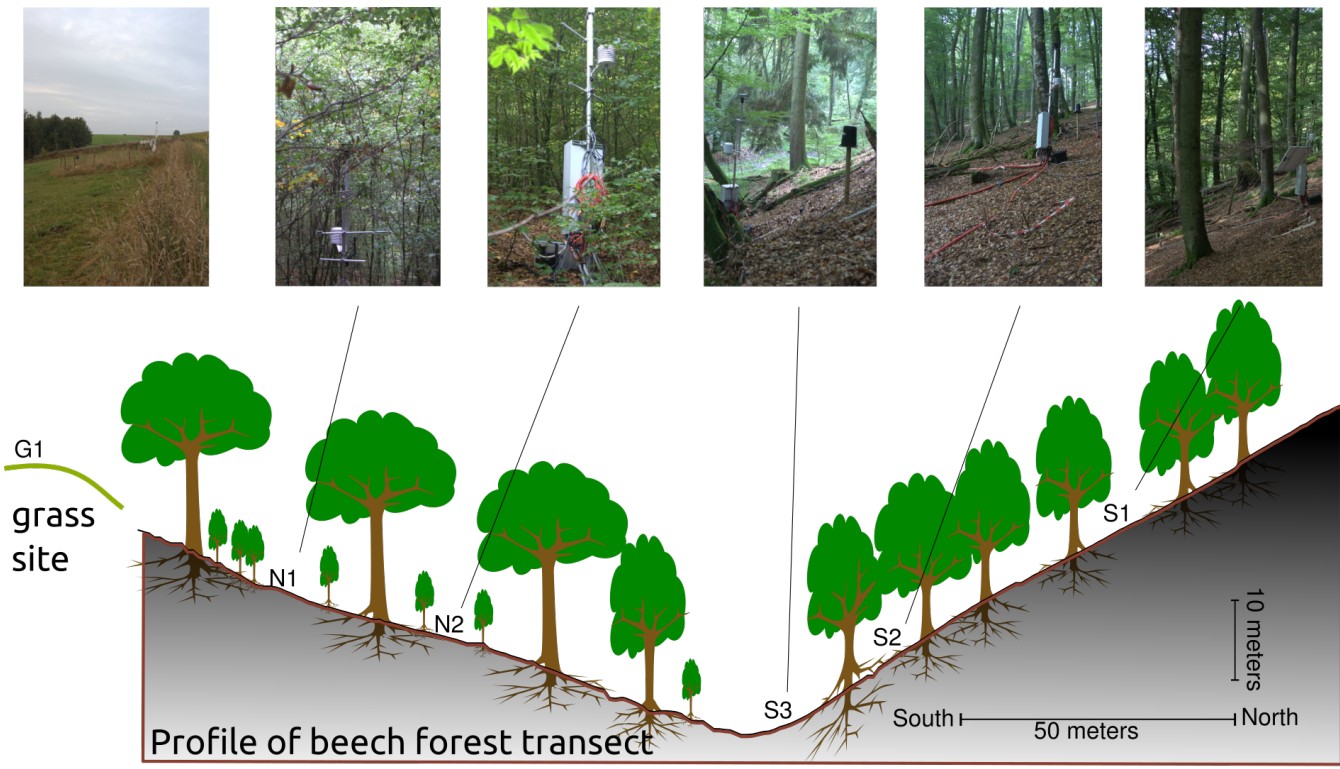

**Figure 1.** Measurement setup along a hillslope transect. Captital letters indicate aspect of the sites (N north-facing, S south-facing) and the numbering represents the position of the site on the hillslope: upslope (1), midslope (2) and downslope (3).

## 2.2 Estimation of atmospheric demand

Our aim is to estimate the potential evaporation of a surface at saturation from first principles and with few, independent input data. Therefore we make use of the concept of thermodynamic limits of convection which was recently established by Kleidon and Renner (2013b) and used successfully to estimate the sensitivity of the hydrologic cycle to global warming (Kleidon and Renner, 2013a; Kleidon et al., 2015) and for grid-scale global predictions of annual average terrestrial evaporation (Kleidon et al., 2014). Here we only illustrate how the concept is used to estimate potential evaporation, for further details the reader is referred to the mentioned publications.

Convection can be thought of as a heat engine which converts a temperature gradient into kinetic energy (Ozawa et al., 2003). To capture the fundamental trade-off of thermodynamic limits of convective exchange, we consider a simple two-box surface-atmosphere system in steady state, which is sketched in Fig. 2. We consider the steady state energy balance of the surface $R_{sn} = J + R_{ln}$. The surface is heated by absorption of incoming solar radiation $R_{sn}$. The turbulent heat fluxes $J$ and the net longwave exchange $R_{ln}$ both cool the surface. The turbulent heat fluxes are comprised by the sensible ($H$) and latent heat flux $\lambda E$: $J = H + \lambda E$. Longwave radiative exchange is represented by a simplified linearized radiation $R_{ln} = k_r(T_s - T_a)$, with




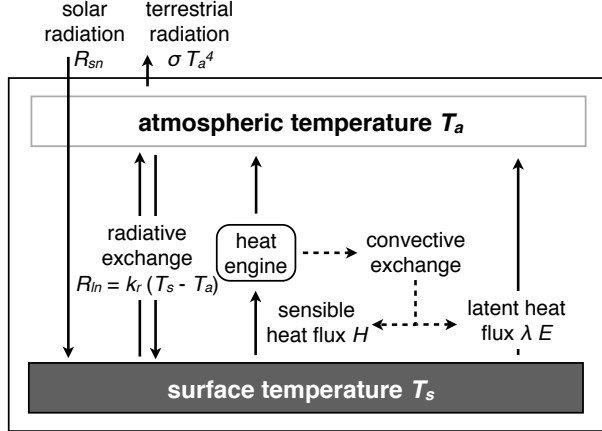

**Figure 2.** Land-atmosphere energy balance scheme for derivation of atmospheric demand adapted after Kleidon et al. (2014).

$T_s$ being the temperature of the surface and $T_a$ the temperature of the atmosphere and $k_r$ being a constant radiative exchange coefficient. The power of the convective heat engine $G$ is fundamentally limited by the Carnot limit:

$$G = J\frac{T_s - T_a}{T_s} \tag{1}$$

5 Different from the classic Carnot engines, the convective heat engine has flexible boundary conditions, namely the temperature gradient $T_s - T_a$ responds to the strength of the convective heat fluxes $J$ through its coupling to the surface energy balance. Rearranging the surface energy balance

$$T_s - T_a = \frac{R_{sn} - J}{k_r} \tag{2}$$

shows that a stronger convective heat flux at a given radiation reduces the temperature gradient. This effectively yields a thermodynamic limit of the maximal power of the convective heat engine. The limit is obtained by inserting this feedback into 10 the Carnot limit Eq. (1) and solving for the maximum with respect to $J$. Hypothesizing that convection actually operates at this limits (subscript $_{opt}$) we obtain a closure for the strength of convective fluxes:

$$J_{opt} = \frac{R_{sn}}{2} \tag{3}$$

Once the convective fluxes are obtained we aim to estimate the partitioning into sensible and latent heat fluxes. Using the Bowen ratio $B$ and bulk formulations for sensible ($H$) and latent heat fluxes ($\lambda E$) we can write:

15 $$B = \frac{H}{\lambda E} = \frac{\rho c_p k_h}{\rho \lambda k_e}\frac{T_s - T_a}{q_s - q_a} = \frac{c_p}{\lambda}\frac{T_s - T_a}{q_s - q_a} \tag{4}$$

with $c_p = 1.2\ JK^{-1}kg^{-1}$ being the heat capacity of air, $\rho = 1\ kgm^{-3}$ the air density, $\lambda = 2.5 \cdot 10^6\ Jkg^{-1}$ the latent heat of vaporisation, $q_s$ and $q_a$ being the specific humidity ($kg \cdot kg^{-1}$) at the surface and the atmosphere respectively. If we assume that the air at the surface and in the atmosphere is saturated with water vapor and that the slope of the saturation vapor pressure curve





($s = \partial e_{sat}/\partial T$) is invariant within this range, the vertical humidity gradient can be written as $q_s - q_a = s(T_s - T_a) \cdot 0.622/p_s$. $e_{sat}$ is the temperature dependent vapor pressure at saturation, $p_s$ the surface pressure ($hPa$) and 0.622 being the ratio of the gas constants for air and water vapor. Introducing the psychrometric constant with $\gamma = c_p/\lambda \cdot p_s/0.622 \approx 65 Pa K^{-1}$ and assuming that the vertical exchange coefficients of air $k_h$ and of vapor $k_e$ are equal, we thus obtain the equilibrium Bowen ratio $B = \frac{\gamma}{s}$ (Schmidt, 1915; Stull, 1988). The equilibrium Bowen ratio depends only on temperature, because $s$ is a non-linear function of temperature and $\gamma$ is approximately constant. We determine $s$ by an empirical equation (Bohren and Albrecht, 1998): $s = s(T) = 6.11 \cdot 5417 \cdot T^{-2} \cdot e^{19.83 - 5417/T}$ with temperature $T(K)$.

Combining the equilibrium Bowen ratio with the concept of maximum convective power we obtain an expression for the potential evaporation, herein referred to as atmospheric demand $E_{opt}$ (Kleidon and Renner, 2013b; Kleidon et al., 2014):

$$E_{opt} = \frac{1}{\lambda} \frac{s}{s + \gamma} \frac{R_{sn}}{2} \qquad (5)$$

Hence, only absorbed solar radiation $R_{sn}$ and temperature data is required to estimate atmospheric evaporative demand, when derived from thermodynamic limits using simplifying assumptions on longwave radiative exchange and the assumption for the equilibrium Bowen ratio.

We also compare our results with the well-established FAO Penman-Monteith grass reference evaporation equation (Allen et al., 1998):

$$E_{PM} = \frac{0.408s(R_n - G) + \gamma \dfrac{900}{T + 273} u VPD}{s + \gamma(1 + 0.34u)} \qquad (6)$$

where $R_n$ is net radiation, $G$ the ground heat flux, $T$ air temperature, $u$ wind speed (both in 2 m height), $VPD$ is the saturation vapor pressure deficit in $kPa$. Equation (6) is a modified Penman-Monteith equation for a standardized grass surface at a daily resolution. As input variables we use the same variables as for $E_{opt}$ (daily average air temperature, shortwave radiation) and in addition net radiation, ground heat flux, relative humidity and wind speed. Because net radiation was not measured at our sites we used an empirical formulation for $R_n$ (Allen et al., 1998, equation (39)). Therefore, data of daily minimum and maximum temperature as well as information on the day of year and latitude was required. The soil heat flux was estimated as $G = 0.1 \cdot R_n$.

## 2.3 Topographic effects on shortwave radiation

In complex terrain the incoming radiation is influenced by the slope and the aspect of the current position (Kondratyev and Fedorova, 1977). To account for these effects we use a topographic radiation correction method which projects the extraterrestrial irradiance on inclined surfaces implemented in the software *r.sun* (Šúri and Hofierka, 2004) which is part of the open-source GRASS GIS platform (http://grass.osgeo.org). *r.sun* estimates potential clear-sky spatial global radiation $R_{g,pot}$ fields for each day of year. To account for cloudy conditions we use the open field site G1 as a reference station and estimate a daily cloud factor $f_c = R_{g,G1}/R_{g,pot}$ using the closest grid point of the respective site. Then $R_{g,pot}$ is multiplied with $f_c$ to obtain a radiation estimate for the sites in the forest $R_{g,c}$. As input for *r.sun* we use a 10 m-resolution DEM (with slope and aspect inputs from





the DEM) and default parameters (Linke atmospheric turbidity coefficient = 2). We thus estimate global radiation on a tilted surface (aspect, slope) and also include topographic shading effects. The effect is largest when the sun angle is low (autumn - winter - spring) and decreases $R_g$ and thus $E_{opt}$ at the north-facing slope while increasing it on the south-facing slope, see Fig. S1. These estimates are generally consistent with radiative observations in sloped terrain (Holst et al., 2005).

## 2.4 Sap flow measurements

The five forested sites are instrumented with multiple sapflow sensors (4 trees at each site, installed between mid of May to November). Heat pulse sensors (East 30 Sensors, Pullman, Washington 99163 USA) based on the heat ratio method (Burgess et al., 2001) have been used as they are less susceptible to natural heating gradients, and require less electrical power (Vandegehuchte and Steppe, 2013). Three needles with equal distance are vertically inserted into the tree. The upper and lower needles measure the change in temperature after a short heat pulse was emitted by a heating element in the middle needle. The heat pulse velocity $v_h$ is proportional to the logarithmic ratio of the temperature differences measured before and at $t_1$ usually 60 sec. after the heat pulse at the the lower ($\Delta T_{dn}$) and upper ($\Delta T_{up}$) needles (Marshall, 1958; Burgess et al., 2001):

$$v_h = \frac{4 k_w t_1 \log \frac{\Delta T_{up}}{\Delta T_{dn}} - x_2^2 + x_1^2}{2 t_1 (x_1 - x_2)} \qquad (7)$$

whereby $k_w$ is the thermal diffusivity of the xylem, and $x_1$ and $x_2$ are the distances to the heater. The sensors have the standard distance of 0.6 cm. This distance can easily be slightly shifted during the installation into living trees, which causes a systematic bias. This sensor alignment bias can be corrected when sap flow is zero (Burgess et al., 2001). Setting $v_h = 0$ and rearranging eq. 7 we can estimate the distance of the upper needle $x_1$ while assuming $x_2 = 0.6$:

$$x_1 = \sqrt{x_2^2 - 4 k_w t_1 \log \frac{\Delta T_{up}}{\Delta T_{dn}}}. \qquad (8)$$

We prefer equation 8 over the published correction in Burgess et al. (2001) because it allows to correct for both, positive and negative nighttime biases in heat velocities. We estimated the corrected distance $x_1$ by assuming zero flow during nighttime between 1 and 4 AM and the median of $x_1$ for the whole installation period. The installation of the sensor needles injures the surrounding xylem vessels and reduces the actual sap flow around the sensor. We applied a polynomial wounding correction $v_c = b V_h + c V_h^2 + d V_h^3$ with wounding correction parameters $b = 1.8558$, $c = -0.0018$, and $d = 0.003$, applicable for a sensor spacing of 0.6 cm and a drilling hole size of 2 mm which is tabulated in Burgess et al. (2001). Finally sap flux density, $SFD$ ($cm^3\, cm^{-2} h^{-1}$), is derived by (Burgess et al., 2001):

$$SFD = v_c \frac{\rho_b (c_w + m_c c_s)}{\rho_s c_s}. \qquad (9)$$

Thus, to estimate the sap flux density, knowledge of xylem wood properties, namely the basic density of wood $\rho_b$, the density of the sap $\rho_s$, the specific heat capacity of the wood matrix $c_w$ and of the sap $c_s$, as well as the water content of the xylem is required. For sap we use the standard parameters for water at 20° C, with $\rho_s = 1\,kg\,cm^3$, $c_s = 4182\,J\,kg^{-1}\,K^{-1}$. The heat capacity of the woody matrix is generally given with $c_w = 1200\,J\,kg^{-1}\,K^{-1}$ (Burgess et al., 2001). The basic density of sapwood





$\rho_b$ measured as dry weight over green volume and the moisture content of the xylem $m_c$ are species-specific parameters. We used $\rho_b = 0.61\,kg\,cm^{-3}$ and $m_c = 0.7$ for the xylem of European beech estimated by Glavac et al. (1990). They sampled about 260 trees at two different sites between 35 and 42 yrs over the course of one year. For the thermal diffusivity of the xylem we used a fixed value of $k_w = 2.5 \cdot 10^{-5}\,m\,s^{-1}$ (Burgess et al., 2001).

The raw measurements obtained every 30 min were quality controlled and suspect data was filtered before further analysis was performed. At three larger trees the outermost sensor readings were replaced by the second sensor depth reading because their annual mean was smaller than the inner sensors which indicates a sensor misplacement into the bark of the tree. The sensors measure the heat pulse velocities at three different radial depths in the tree. The daily mean sap flux density per tree is obtained by an average of all readings per depth and day. Units of $SFD$ were converted to $m^3\,m^{-2}\,d^{-1}$. A arithmetic mean of
the tree-average sap flux density was used to obtain the site-average sap flux density.

## 2.5    Biometric measurements

A forest inventory for all sites was done in March 2012. The circumference at breast height of all trees with circumference larger or equal 4 cm was measured in a 20 m by 20 m plot for each site. Stem basal area was calculated and a total stand basal area was computed for each site.

Leaf area index was measured with a LICOR LAI-2200 at all forested sites in two campaigns. The summer campaign was carried out on 2012-08-11 and the winter campaign on 2014-03-20. Here we use the measurements taken with all rings of the LAI-2200. The LAI was averaged from 36 measurements points per site. The difference between summer and winter LAI should reveal the actual leaf area index without stems and topographic shading effects.

## 2.6    Upscaling of sap flow to tree and stand transpiration

Measurements of sap flux density only provide a relative measure of the velocity of the ascending xylem sap. To obtain tree water fluxes a representative xylem area per sensor depth is assumed which is then multiplied with the corresponding sap flux density (Burgess et al., 2001). The needles measure heat-pulse velocities at depths of 5, 18, and 30 mm within the stem. Following the manufacturer manual we assigned for each sensor depth an representative radius of 15, 25, and 40 mm below the cambium radius $r_x$, which is obtained by assuming that the cambium takes 1% of the total radius, thus $r_x = 0.99 \cdot DBH/2$.

Tall trees with a diffuse-porous xylem structure such as beech are known to have a large sapwood to basal area ratio (Köstner et al., 1998). Further, the radial profile of the sap flux density varies in these trees. Therefore deep measurements are ideally required (Gebauer et al., 2008). In the absence of these deeper measurements we follow the reasoning of Lüttschwager and Remus (2007) to derive uncertainty ranges for the inner conducting sapwood area. A minimal estimate is obtained by assuming a representative annulus of 15 mm depth for the sensor and zero flow in the inner sapwood, which is calculated by $A_{S3,min}$.
Assuming that sap flux density remains constant at the innermost sensor level throughout the inner sapwood area provides a maximal estimate of sap flux. Most realistic is the assumption of a linear decline reaching no flow at the estimated heartwood




radius $A_{S3,lin}$ similar to Pausch et al. (2000):

$$A_{S3,min} = \pi \left( (r_x - 25mm)^2 - (r_x - 40mm)^2 \right) \tag{10}$$

$$A_{S3,lin} = \pi((r_x - 25mm)^2 - \frac{1}{3}((r_x - 25mm)^2 + (r_x - 25mm)r_h + r_h^2)) \tag{11}$$

$$A_{S3,max} = \pi \left( (r_x - 25mm)^2 - (r_x - r_h)^2 \right) \tag{12}$$

The radius of the heartwood $r_h$ is obtained from sapwood area estimates, explained below. For small trees with radius smaller than the representative annulus depth we set the area to 0.

To upscale to the site level we use the inventory data which provides the number of trees in the stand, DBH and species information. Stem sapwood area was computed using published allometric power law relationships of the form $A_s = aDBH^b$ based on diameter at breast height (DBH). For beech and alder we used the relationship published in Gebauer et al. (2008), for

oak we used Schmidt (2007) and Alsheimer et al. (1998) for spruce.

Daily sap flux density per sensor depth was averaged for each species and tree status (dominant vs. suppressed) for each site. Missing sap flux density data was filled by linear regression with neighboring sites. Largest data gaps were filled at sites N2 and S2. Finally, the upscaled daily stand transpiration was obtained by summing up the product of sap flux density per depth $D$ and sap wood area per depth $A_{sap}(D)$ for all trees and dividing by the area of the inventory $A_{stand}$:

$$E_{sap} = \frac{1}{A_{stand}} \sum_{tree=1}^{n} \sum_{D=1}^{3} (SFD(D) \, A_{sap}(D)). \tag{13}$$

## 2.7   Root water uptake estimation

As another means to estimate transpiration we use soil moisture measurements. Soil moisture sensors (Decagon 5TE soil moisture sensors, with an accuracy of $\pm 3\%$ volumetric water content and resolution of 0.08%) are installed at 3 profiles at each site at 10 - 30 - 50 cm depth and one deeper sensor (approx. 70 cm) at one of the profiles. There is a range of methods

to estimate root water uptake from soil moisture observations (Shuttleworth, 1993; Cuenca et al., 1997; Wilson et al., 2001; Schwärzel et al., 2009; Breña Naranjo et al., 2011; Guderle and Hildebrandt, 2015). Here, we employ a simplified soil water budget method. The soil water continuity equation at a point in the soil may be written as (Cuenca et al., 1997):

$$\frac{\partial \theta}{\partial t} = -\frac{\partial q_i}{\partial x_i} + S \tag{14}$$

where $\theta$ is the soil moisture content, $q$ summarizes any soil water fluxes over an orthogonal coordinate system $x_i$ with horizontal

$(x, y)$ and vertical directions $(z)$. The sink term $S$ is the root water uptake, which when integrated over the soil volume $V_s$ yields the transpiration flux $E_{RWU}$ (Cuenca et al., 1997):

$$E_{RWU} = \int\limits_{V_s} S \, dV_s = \int\limits_{V_s} \frac{\partial \theta}{\partial t} dV_s + \int\limits_{V_s} \frac{\partial q_i}{\partial x_i} dV_s \tag{15}$$

Hence, to estimate $E_{RWU}$ we need to resolve the two terms in equation 15. The first term is the temporal evolution of soil moisture, which is in principle measured by the soil moisture sensors. The second term describes soil water fluxes within the





soil such as downward fluxes during an infiltration event. The soil water fluxes $q_i$ themselves depend on soil moisture, through its role in determining soil water potentials and unsaturated hydraulic conductivity. Soils along hillslopes have large spatial heterogeneity of their hydraulic properties and pose many influences on the second term of eq. (15), which makes continuous estimation of root water uptake difficult.

There are, however, periods and locations where soil water fluxes $q_i$, such as drainage, capillary rise etc, are of minor importance. Especially during dry conditions the reduction of soil moisture is dominated by root water uptake and soil evaporation during daytime, as illustrated in Fig. 3. Under these radiation-driven conditions, we observe rather constant nighttime moisture levels which thus indicates that soil water fluxes are not active and nocturnal root-soil exchange is negligible. Thus, night time soil moisture dynamics are a practical filter to exclude days with dominant drainage or capillary rise fluxes.

To estimate daily root water uptake we use soil moisture observations on a half-hourly basis. We first quantify the day-time change in soil moisture per sensor $\Delta\theta_{d,s}$ by cumulative sum of differences of soil moisture (at 30 min intervals) from astronomical sunrise to sunset:

$$\Delta\theta_{d,s} = \sum_{t=t_{sunrise}}^{t=t_{sunset}} (\theta_{t+1} - \theta_t) \tag{16}$$

Assuming that the sensor observation is representative for the respective soil depth $\Delta z$ of the sensor, we obtain a flux estimate

$E_{RWU,d,z} = -\Delta\theta_{d,s}\Delta z$ per soil layer. A constant soil depth of $\Delta z = 200~mm$ per sensor level was assumed for all soil profiles. The sum of all $E_{RWU,d,z}$ per profile then gives a total daily root water uptake $E_{RWU,d}$ per profile.

To filter for drainage and other soil water fluxes we then filtered the data for (i) precipitation (daily sum $< 0.1mm$ and rainfall of previous day $< 1mm$), (ii) only negative daytime soil moisture changes $\Delta\theta_{d,s}$ and (iii) absolute cumulative night-time changes in soil moisture $|\Delta\theta_{n,s}| < 0.1Vol\%$. Approximately 30% of the actual data were retained, with details listed in

Table S2.

Generally, diurnal variations of soil moisture are large in the upper soil depths compared to deeper layers. Thus for estimating the total profile root water uptake, the upper sensors are very important. Unfortunately we had to face sensor failure of top soil sensors at site N1, profile 2 and site S1 profiles 2 and 3, see Table S2 for an overview of available data. At these soil profiles we used the root water uptake estimates from the 2nd sensor level to fill the missing data. Two profiles, one at G1 and another

at N2 were completely disregarded for site averaging due to sensor failures. The site-average was obtained from at least 2 profile-based $E_{RWU}$ estimates. At the grass site this condition was relaxed because there were only a few days where $E_{RWU}$ could be estimated from both profiles. At these few days the estimates of the two profiles were comparable.

## 2.8  Statistical analyses

To test if the transpiration estimates are driven by potential atmospheric demand we use linear ordinary least squares (OLS)

regression with the transpiration estimates as response variable and $E_{opt}$ as independent variable. The slope of the regression with sap flux density is denoted by $b_{SFD} = \frac{dv_{sap}}{dE_{opt}}$ with units $[m^3\,m^{-2}d^{-1}]/[mm\,d^{-1}]$. For $E_{sap}$ the slope is denoted $b_{sap} = \frac{dE_{sap}}{dE_{opt}}$ and for $E_{RWU}$ the slope is denoted $b_{RWU} = \frac{dE_{RWU}}{dE_{opt}}$. The latter two regression slopes have non-dimensional units $[mm\,d^{-1}]/[mm\,d^{-1}]$.





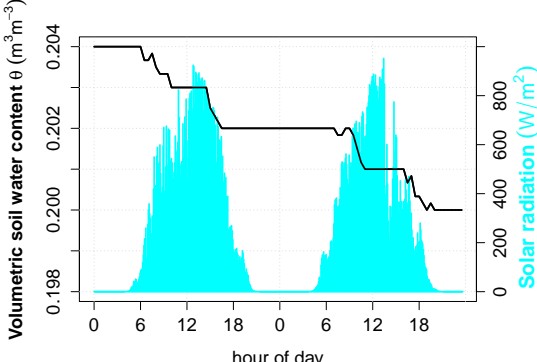

**Figure 3.** Observed diurnal decline in soil moisture over two sunny days in summer 2013 to illustrate the approach to estimate daily root water uptake from diurnal soil moisture variations. $E_{RWU}$ is estimated at sensor level per day as a cumulative sum of differences from sunrise to sunset, in the absence of any nighttime variations or daytime increases in soil moisture.

To estimate the potential influence of other independent variables on transpiration, such as soil moisture or vapor pressure deficit, we use the residuals of the OLS regression as response variable of a further linear regression. The explanatory power of the other variable is reported by the adjusted explained variance (denoted by $R^2_\theta$ for soil moisture effects) which is a diagnostic of the OLS regression.

Note that by using time series of daily data, which are shaped by the seasonal cycle, the assumption of independence of the predictor variables in the linear regression is not often justified. The effect of serial dependence generally does not bias the regression coefficients, but reduces the statistical significance of a regression. Therefore, we estimate the standard deviation of the regression coefficient ($\sigma_{slope}$) and its reported significance with a pre-whitening procedure of Newey and West (1994) provided by function *coeftest* of the *R* packages lmtest (Zeileis and Hothorn, 2002) and sandwich (Zeileis, 2004). All data
analysis was done in *R* (R Core Team, 2014) and the R-package data.table (Dowle et al., 2014).

## 3    Results

### 3.1    Meteorological observations in 2013

Daily time series of temperature, global radiation, precipitation and site-average soil moisture content for 2013 are shown in Fig. 4. The average annual (growing season, May - October) temperature was 7.9 (13.7) $^\circ C$, with an average global radiation
of 117 (169) $W/m^2$. Highest temperatures were observed in a short period in June and a longer period between mid July and August. We observed a total annual precipitation of 866 $mm$ (no catch correction etc. applied), which was, however, unevenly distributed throughout the year. Most precipitation fell in June (110 $mm$), with low rainfall during July and August (34 and 38 $mm$ respectively) and more rainfall in autumn months. Hence, the soil moisture recession lasted from July to September (bottom grey line in Fig. 4).





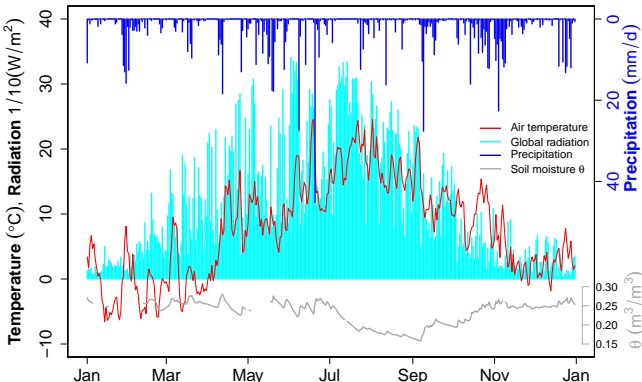

**Figure 4.** Meteorological forcing and average soil moisture at the grassland site (G1) for 2013. A few data gaps were filled with data from nearby grassland sites.

## 3.2 Site-scale transpiration estimates

Two independent site-scale transpiration estimates are evaluated in the following. The first estimate, $E_{sap}$, was obtained by upscaling sap flow measurements with the forest plot inventory data. The second estimate was derived by averaging $E_{RWU}$ of the 2-3 soil profiles per site. Both transpiration estimates are shown as daily time series for site N1 at the north- and S1 at the south-facing hillslope in Figure 5. The north-facing sites have lower potential evaporative demand (estimated by topographically corrected incoming solar radiation) than the south-facing sites. Variations of daily soil water contents are quite similar between the sites. However, the north-facing site is on average slightly wetter ($\overline{\theta} = 0.21 \pm 0.04$) than the south-facing site ($\overline{\theta} = 0.18 \pm 0.03$). There is a large temporal variability in the atmospheric forcing represented by $E_{opt}$ which is also found for both transpiration estimates. Highest transpiration rates are seen in the dry and sunny period in July. Comparing the sites at the different hillslopes we find similar magnitudes of $E_{sap}$. However, $E_{RWU}$ is found to be much higher than $E_{sap}$ in spring and early summer at the north-facing sites.

Seasonal totals of $E_{sap}$ and potential evaporation are reported in Table 1. The $E_{sap}$ estimates for the upper north and south-facing hillslope sites are relatively similar. Highest $E_{sap}$ is estimated at the valley bottom site S3, which amounts to a difference of about 50 mm for the seasonal total. Large uncertainty in the nominal value of $E_{sap}$ is due to the assumed depth of the sapwood which is estimated from data of the innermost sensor and allometric relationships of sapwood. Thereby the range between different conducting sapwood depths is largest at the valley site because there are more tall trees where this uncertainty matters.

## 3.3 Transpiration response to atmospheric demand

To test if the transpiration estimates are driven by potential atmospheric demand we performed a linear regression of the site-average values with $E_{opt}$ as independent variable. We found that $E_{sap}$ was relatively low in May and early June after leaf out and after mid-October with leaf senescence. In order to avoid phenological effects in the statistical analysis, we restrict the



**Table 1.** Seasonal totals in $mm$ of estimated stand transpiration $E_{sap}$ and potential evaporation $E_{opt}$ and $E_{PM}$. For the totals we used the period 2013-05-12 - 2013-10-31. For $E_{sap}$ we present a range of estimates with minimal, linear decline and maximal estimate with respect to the depth of conducting inner sapwood (Eqn. 10, 11,12).

| site | $E_{sap}$ | $E_{opt}$ | $E_{PM}$ |
|------|-----------|-----------|----------|
|      | min - lin - max |    |    |
| N1 | 106 - 128 - 156 | 240 | 351 |
| N2 | 76 - 115 - 164  | 232 | 345 |
| S3 | 127 - 176 - 244 | 296 | 387 |
| S2 | 104 - 129 - 170 | 293 | 386 |
| S1 | 96 - 122 - 160  | 295 | 387 |
| G1 |                 | 270 | 371 |

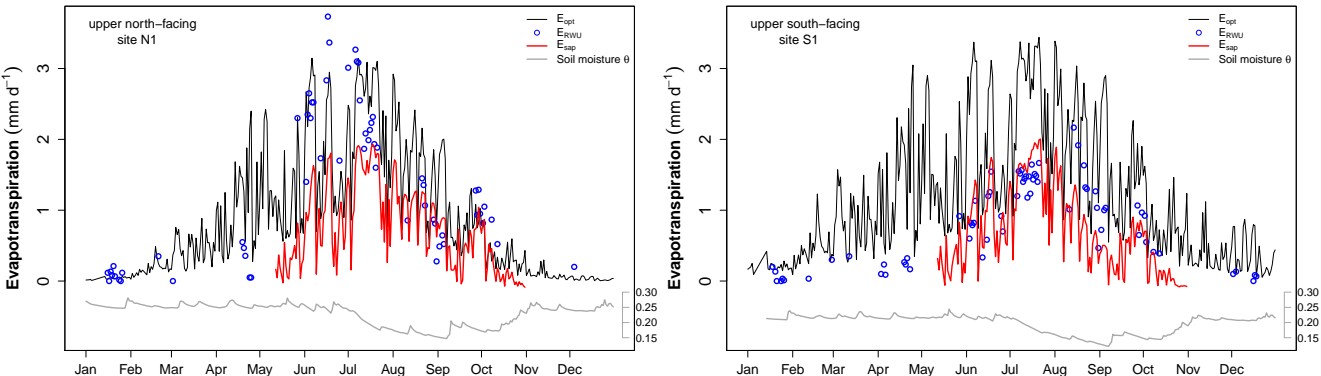

**Figure 5.** Time series of evapotranspiration estimates for north-facing hillslope site N1 in the left panel and south-facing site S1 in the right panel. $E_{opt}$ is the potential evaporation derived from maximum power, $E_{RWU}$ (blue points) and $E_{sap}$ are transpiration estimates.

following analyses to the vegetation period from 10th of June to 20th of October. Respective scatterplots are shown in Figure 6 and a summary of the results is reported in Table 2.

Figure 6 shows that both observations almost linearly increase with $E_{opt}$. For $E_{sap}$ we find very high linear correlations with an average of $r^2_{E_{opt}} = 0.82$, (Table 2). The relatively large temporal variability in $E_{sap}$ seen in Fig. 5 with a coefficient of variation $CV = 0.68$ across all forest sites is explained by the atmospheric demand $E_{opt}$ ($CV = 0.53$). It is interesting to note that across all forest sites global radiation has only a slightly lower average $r^2 = 0.79$, whereas air temperature with $r^2 = 0.55$ and vapor pressure deficit with $r^2 = 0.66$ have less explanatory power. The correlation of site-average $E_{RWU}$ to $E_{opt}$ is much lower ($r^2 = 0.49$, averaged among all sites) than the correlation of $E_{sap}$ to $E_{opt}$. All meteorological variables show lower correlations to $E_{RWU}$ than to $E_{sap}$, which indicates a larger uncertainty of the soil moisture derived transpiration estimate.

We also tested if the regression residuals show correlations with other daily observations. The reported adjusted squared correlation of a linear regression of the residual with VPD, wind speed and soil moisture at the site level is reported in Table 2.





**Figure 6.** Site-average $E_{RWU}$ (blue circles) and $E_{sap}$ (red triangles) as function of $E_{opt}$ during vegetation period (10.June - 20.October) in 2013. Each panel presents data of one site (see Figure 1 for site identifiers). The size of the symbols corresponds to the daily average soil moisture content. The histogram shows the distribution of site-average soil moisture. Solid lines depict the linear regression as tabulated in Table 2.




**Table 2.** Regression statistics for $E_{sap}$ and $E_{RWU}$ to $E_{opt}$, with $n$ providing the number of observations (days). The slope and intercept are reported with the estimated standard deviation of the coefficients with $\pm\sigma$. Significance of the coefficients is indicated by stars: $p < .001$, ***; $p < .01$, **; $p < .05$, * . $r^2_{E_{opt}}$ and $r^2_{E_{PM}}$ are the linear squared correlation coefficients to $E_{opt}$ and $E_{PM}$, respectively. The last three columns report the adjusted explained variance of a linear regression of the regression residuals for daily average vapor pressure deficit ($R^2_{VPD}$), daily average wind speed ($R^2_u$), and daily site-average volumetric water content ($R^2_\theta$).

| variable | site | n | slope | intercept | $r^2_{E_{opt}}$ | $r^2_{E_{PM}}$ | $R^2_{VPD}$ | $R^2_u$ | $R^2_\theta$ |
|---|---|---|---|---|---|---|---|---|---|
| | N1 | 130 | $0.61 \pm 0.02$ *** | $-0.03 \pm 0.05$ | 0.88 | 0.89 | 0.01 | 0.00 | 0.02 |
| | N2 | 129 | $0.57 \pm 0.03$ *** | $-0.03 \pm 0.05$ | 0.88 | 0.89 | 0.01 | 0.01 | -0.00 |
| $E_{sap}$ | S3 | 130 | $0.84 \pm 0.08$ *** | $-0.36 \pm 0.10$ *** | 0.83 | 0.87 | -0.01 | 0.01 | 0.19 ** |
| | S2 | 109 | $0.62 \pm 0.07$ *** | $-0.27 \pm 0.09$ ** | 0.75 | 0.80 | -0.01 | 0.01 | 0.25 * |
| | S1 | 130 | $0.58 \pm 0.07$ *** | $-0.23 \pm 0.07$ ** | 0.78 | 0.83 | -0.01 | 0.00 | 0.24 ** |
| | N1 | 36 | $0.77 \pm 0.18$ *** | $0.19 \pm 0.27$ | 0.52 | 0.50 | 0.04 | 0.03 | 0.64 *** |
| | N2 | 30 | $0.76 \pm 0.19$ *** | $0.20 \pm 0.40$ | 0.67 | 0.69 | -0.02 | 0.10 | 0.11 |
| $E_{RWU}$ | S3 | 35 | $0.56 \pm 0.11$ *** | $-0.21 \pm 0.22$ | 0.44 | 0.55 | -0.03 | 0.32 ** | 0.53 *** |
| | S2 | 30 | $0.52 \pm 0.11$ *** | $0.11 \pm 0.19$ | 0.47 | 0.58 | -0.02 | 0.12 | 0.45 |
| | S1 | 39 | $0.44 \pm 0.03$ *** | $0.13 \pm 0.06$ * | 0.63 | 0.60 | -0.03 | 0.03 | -0.02 |
| | G1 | 28 | $0.75 \pm 0.34$ * | $0.32 \pm 0.77$ | 0.22 | 0.21 | 0.04 | -0.04 | 0.63 *** |

Vapor pressure deficit does not explain residual variance of $E_{sap}$ or $E_{RWU}$. Also wind speed does not show residual correlation for $E_{sap}$ but higher positive residual correlation is found for $E_{RWU}$ at the lower sites N2, S2 and significant at S3. Both of these variables are used as external forcing in the Penman equation. Therefore we calculated the squared correlation also for the FAO Penman-Monteith reference evaporation $E_{PM}$, Eqn (6). One can see from Table 2 that $r^2_{E_{PM}}$ is fairly similar, follows the same pattern, and is on average only slightly larger ($r^2_{E_{PM}}$ =0.86) than the correlation of $E_{sap}$ to $E_{opt}$. Note that $E_{PM}$ is on average 1.44 times larger than $E_{opt}$ with a correlation between $E_{PM}$ and $E_{opt}$ of $r^2 = 0.98$.

In contrast to the meteorological variables we found that the residuals are significantly correlated to the site-average soil moisture content at some sites. $E_{sap}$ is significantly affected by soil moisture deficits at the south-facing sites. At these sites we find significant residual correlation of soil moisture $0.19 < R^2_\theta < 0.25$, which results in lower correlation to $E_{opt}$ as compared to the two north-facing sites. Further, we find significant negative intercepts in the $E_{sap}$–$E_{opt}$ relationship. Even more affected by soil water limitation is $E_{RWU}$ with significant values of $R^2_\theta > 0.5$ at N1, S3, and G1. However, at S1 there is no residual correlation at the site-average level, but this is probably an effect of deriving a representative site-average from $E_{RWU}$ profile estimates, because one soil profile at S1 actually shows a significant residual correlation (Table S3). The potential effect of soil moisture on transpiration is visualized by the size of the symbols in Fig. 6. Thereby, $E_{RWU}$ tends to be above the regression line under moist and warm conditions in early summer particularly at sites N1 and G1.

Figure 6 and the regression statistics allow for a further comparison of $E_{sap}$ and $E_{RWU}$. Generally there is a good agreement of both estimates with $0.5 < r^2 < 0.9$, but there are site-specific deviations between the site transpiration estimates. The upper





south-facing sites S1 and S2 agree well in magnitude, while $E_{sap}$ for site S3 appears to have an offset of about $0.5 mm\,d^{-1}$. This offset could be influenced by the apparent site heterogeneity. The site is situated in the transition between the steep hillslope and the riparian zone of the nearby creek. The $E_{sap}$ upscaling represents this transition zone by the 20 m by 20 m forest inventory size and assumes constant sapwood properties for all trees within this site. On the other hand three soil moisture

profiles may be too few in this heterogeneous zone which complicates the estimation of a representative site-average $E_{RWU}$ value. At the north-facing sites we find that $E_{sap}$ is lower than the $E_{RWU}$ estimates. Especially during early summer when soil moisture was still high $E_{RWU}$ was found to be almost twice as large as $E_{sap}$ and of similar magnitude as $E_{opt}$ at these sites. However, there was a better agreement in late summer, when soil moisture declined, see Figure 5. This indicates that the root water uptake estimates tend to be higher under moist conditions which is reflected by the positive residual correlation to soil

moisture reported in Tables 2 and S3.

### 3.3.1    Response at tree level

By using tree-averaged sap flow density ($SFD$) measurements directly we can assess how strong single trees respond to $E_{opt}$. Results of a linear regression analysis of daily tree average $SFD$ to $E_{opt}$ are reported in Table 3 (see also Figure S3). $SFD$ of every tree shows a strong linear relation to $E_{opt}$, with varying slopes and intercept terms. Because most of the variability

is explained by $E_{opt}$, we can simply compare the slope of the regression $b_{SFD}$. First, we find that $b_{SFD}$ is especially high at the tall beech trees at the north-facing sites (1.26 - 1.64 $[m^3\,m^{-2}d^{-1}]/[mm\,d^{-1}]$), Table 3). These sampled tall trees at the north-facing sites are dominant trees without other tall trees in their vicinity. All other trees show much smaller values of $b_{SFD}$ (0.26 - 0.88 $[m^3\,m^{-2}d^{-1}]/[mm\,d^{-1}]$). Secondly, we find that $b_{SFD}$ is on average larger at the north- than at the south-facing trees. Last, we also find that the oak tree at site S1 has the lowest $b_{SFD}$. For these reasons the upscaling of $SFD$ to $E_{sap}$ was

done for each site separately and distinction was made between dominant and small trees as well as between species.

The residual regression analysis shows a similar pattern as the site-average values, with low influence of wind speed and VPD. Site-average soil moisture content ($\theta$) shows significant effects at 9 out of 14 trees, see Table 3. Thereby all beech trees at the south-facing sites show an influence of $\theta$ on the sap flow residuals, with a maximum adjusted explained variance of $R_\theta^2 = 0.4$ at the upper south-facing site S1. In addition, the 3 of 4 small beech trees at the north-facing sites show significant

influences of $\theta$.

### 3.3.2    Soil profile response and root water uptake profiles

Generally, the measurement setup allows to differentiate between three different levels of aggregation of root water uptake estimates: site-average, per profile, and per depth of the sensor.

Root water uptake per profile is obtained by summing up the estimates of each sensor. We find that 15 out of 16 profiles

show significant slopes $b_{RWU}$ (Table S3) which reveals a significant linear relationship with $E_{opt}$. The profile-based results also highlight that there is considerable within-site variability of the response of $E_{RWU}$ to $E_{opt}$, with significant slopes ranging between 0.15 and 1.17. Testing the regression residuals for an influence of soil moisture content we find 10 of 16 forest soil profiles with significant residual correlation $R_\theta^2$. At the grass site two profiles showed even larger residual correlations to $\theta$





**Table 3.** Regression statistics of daily $SFD$ as average per tree and per site as extra row. Column $DBH$ is the diameter at breast height in $cm$. $n$ is the number of observations (days). The slope $b_{SFD}$ and intercept of the linear univariate regression of $SFD$ to $E_{opt}$ with $\pm\sigma$ reporting the estimated standard deviation of the coefficients. Significance of the coefficients is indicated by stars: $p < .001$, ***; $p < .01$, **; $p < .05$, *. $r^2_{E_{opt}}$ and $r^2_{E_{PM}}$ are the linear squared correlation coefficients of $SFD$ to $E_{opt}$ and $E_{PM}$, respectively. The last three columns report the adjusted explained variance of a linear regression of the regression model residuals for the variables: vapor pressure deficit ($R^2_{VPD}$), wind speed ($R^2_u$), and site-average volumetric water content ($R^2_\theta$).

| site | tree | species | DBH | n | $b_{SFD}$ | intercept | $r^2_{E_{opt}}$ | $r^2_{E_{PM}}$ | $R^2_{VPD}$ | $R^2_u$ | $R^2_\theta$ |
|------|------|---------|-----|-----|-----------|-----------|-----------------|----------------|-------------|---------|--------------|
| N1 | 1 | beech | 66 | 130 | $1.30 \pm 0.05$ *** | $0.09 \pm 0.13$ | 0.88 | 0.88 | 0.03 | 0.07 ** | 0.11 ** |
| N1 | 2 | beech | 58 | 130 | $1.26 \pm 0.04$ *** | $-0.06 \pm 0.08$ | 0.92 | 0.84 | 0.00 | 0.03 * | -0.01 |
| N1 | 3 | beech | 9 | 130 | $0.88 \pm 0.05$ *** | $-0.08 \pm 0.09$ | 0.82 | 0.89 | 0.03 | -0.00 | 0.08 * |
| N1 | 4 | beech | 8 | 130 | $0.63 \pm 0.08$ *** | $-0.22 \pm 0.08$ ** | 0.79 | 0.88 | -0.00 | 0.02 | 0.29 *** |
| N1 | | | | 130 | $1.02 \pm 0.03$ *** | $-0.07 \pm 0.07$ | 0.90 | 0.91 | 0.01 | 0.01 | 0.00 |
| N2 | 1 | beech | 53 | 97 | $1.31 \pm 0.08$ *** | $0.23 \pm 0.16$ | 0.83 | 0.83 | -0.01 | -0.00 | 0.03 |
| N2 | 2 | beech | 49 | 97 | $1.64 \pm 0.09$ *** | $-0.06 \pm 0.18$ | 0.87 | 0.88 | 0.00 | -0.00 | -0.01 |
| N2 | 3 | beech | 10 | 97 | $0.68 \pm 0.04$ *** | $-0.11 \pm 0.05$ * | 0.89 | 0.82 | 0.01 | 0.00 | -0.00 |
| N2 | 4 | beech | 8 | 97 | $0.51 \pm 0.08$ *** | $-0.14 \pm 0.08$ | 0.80 | 0.78 | -0.01 | 0.00 | 0.26 ** |
| N2 | | | | 97 | $1.04 \pm 0.06$ *** | $-0.02 \pm 0.11$ | 0.87 | 0.88 | -0.00 | -0.01 | 0.02 |
| S1 | 1 | beech | 43 | 130 | $0.81 \pm 0.08$ *** | $-0.23 \pm 0.10$ * | 0.77 | 0.80 | -0.01 | 0.01 | 0.34 ** |
| S1 | 2 | oak | 40 | 130 | $0.26 \pm 0.01$ *** | $-0.10 \pm 0.02$ *** | 0.92 | 0.94 | 0.05 * | 0.01 | -0.01 |
| S1 | 3 | beech | 39 | 97 | $0.39 \pm 0.12$ ** | $-0.14 \pm 0.12$ | 0.64 | 0.81 | 0.01 | 0.01 | 0.40 *** |
| S1 | 4 | beech | 46 | 130 | $0.40 \pm 0.05$ *** | $-0.13 \pm 0.06$ * | 0.76 | 0.87 | -0.01 | 0.01 | 0.40 *** |
| S1 | | | | 130 | $0.49 \pm 0.05$ *** | $-0.17 \pm 0.06$ ** | 0.80 | 0.84 | -0.01 | 0.00 | 0.30 ** |
| S3 | 1 | beech | 45 | 124 | $0.52 \pm 0.06$ *** | $-0.22 \pm 0.07$ *** | 0.83 | 0.86 | -0.01 | 0.02 | 0.22 *** |
| S3 | 2 | beech | 39 | 124 | $0.86 \pm 0.08$ *** | $-0.35 \pm 0.10$ *** | 0.84 | 0.90 | -0.00 | -0.00 | 0.09 * |
| S3 | | | | 124 | $0.69 \pm 0.07$ *** | $-0.28 \pm 0.08$ *** | 0.84 | 0.87 | -0.01 | 0.01 | 0.12 * |

than to $E_{opt}$ ($R^2_\theta = 0.71, 0.72$). Testing for additional correlation of VPD and wind speed on the regression residuals of $E_{RWU}$ to $E_{opt}$ we find that VPD has generally low additional value with only three profiles with significant influences. Windspeed shows significant residual correlation at 3 out of 16 profiles.

Root water uptake at a specific sensor level ($E_{RWU,d,z}$) allows to assess at which depths plant water extraction and/or capillary rise due to soil evaporation are effective. $E_{RWU}$ is detected at almost all soil moisture sensors during the growing season in 2013 (Fig. S4). However, $E_{RWU}$ at the deepest sensor is much lower than what is observed at the top (0 - 20 cm) soil layer. Comparing the grass site (G1) with the forest sites we find that the grass site has much larger $E_{RWU}$ from the top-layer (0 - 20cm) than from deeper layers. Further, we can see that the top-layer uptake decreases during the drier summer period July to September. The reduction is strongest at the grass site, whereas the forest sites show a more evenly distributed uptake with soil depth.





**Table 4.** Site topographic characteristics with average inclination of each site is given in column 'slope angle'. Column, $R_{g,c}$ is the average global radiation $R_{g,c}$, and $E_{opt}$ and $E_{PM}$ are all averaged for the vegetation period 10.06–20.10. Values of site G1 are measured directly, values of the forested sites are estimated by the topographic correction of solar radiation.

| site | slope angle | $R_{g,c}$ $(Wm^{-2})$ | $E_{opt}$ $(mm\,d^{-1})$ | $E_{PM}$ $(mm\,d^{-1})$ |
|------|-------------|--------------|-----------------|----------------|
| N1 | 14° | 153 | 1.46 | 2.19 |
| N2 | 18° | 148 | 1.41 | 2.15 |
| S3 | 30° | 192 | 1.82 | 2.44 |
| S2 | 22° | 190 | 1.80 | 2.43 |
| S1 | 24° | 191 | 1.81 | 2.44 |
| G1 | 8° | 173 | 1.65 | 2.33 |

## 3.4 Influence of topography and stand structure

The results show that most of the daily variability in both transpiration estimates is driven by atmospheric demand. The slope of the linear regression of $E_{sap}$, $SFD$ and $E_{RWU}$ to $E_{opt}$ provides a summary statistic of how strong a site or a tree responds to atmospheric demand. In the following we evaluate which location factors correlate with the slopes of the linear regressions. We concentrate on topographic and stand structural parameters.

The most obvious factor is the aspect of the sites. While $b_{sap}$ does not show aspect related differences, we find that $b_{SFD}$ and $b_{RWU}$ are higher at the north-facing sites and lower at the south-facing sites. This is found at the site-average level (Table 2), at the soil profile level (Table S3) and at the tree level (Table 3). The differences are, however, only significant for $b_{SFD}$ at the tree level being larger at the north-facing slope with average $b_{SFD} = 1.03$ than at the south-facing slope $b_{SFD} = 0.63$, which is significant at the 0.05 level with a Student's t-test.

Another topographic factor influencing the response to atmospheric demand is the inclination of the sites, where we find that the steeper the site the lower the slope of the linear regression of $SFD$ to $E_{opt}$, compare Tables 3 and 4. Both aspect and inclination affect the received solar radiation. According to the topographic correction of solar radiation, the north-facing sites received about 85% to 89% of the solar radiation which was observed at the grass site. In contrast the south-facing sites received about 108-110%, see Table 4. This difference is confirmed by observations of air temperature within the forest, which reveal the same ranking across the sites as $R_{g,c}$.

Apart from the topographic data, the stand structure varies remarkably (see Table 5). At the two north-facing sites we find much understorey ($> 98\%$ of trees smaller than DBH 15cm) and a few tall trees, but no medium-sized trees between 15 and 50 cm DBH. The tall, dominant trees therefore have a well-lit canopy. The taller young trees form a secondary, lower but closed canopy layer. In contrast the south-facing sites S1 and S2 have about 20 medium sized trees per 20 by 20 m plot mainly between 16 and 65 cm DBH. These sites form a rather closed one-layer forest canopy. The valley site shows the largest tree size diversity with small, medium and very tall trees. The stand composition differences of the sites result in quite different



**Table 5.** Observed biometric characteristics of the 5 forested sites along the transect. Column names: $n_{tree}$ ... number of trees with circumference $> 4\,cm$ in a 20 by 20 m reference plot, diameter at breast height (DBH) distribution with minimum, first quartile, mean, third quartile and maximum $(cm)$, total stand basal area $A_b$ $(m^2ha^{-1})$, total stand sapwood area estimated from published allometric relationships $A_s$ $(m^2ha^{-1})$, see section 2.5. Site leaf area index (LAI) was derived from the difference of summer $LAI_s$ and winter $LAI_w$.

| site | $n_{tree}$ | DBH (cm) | | | | | A ($m^2ha^{-1}$) | | LAI ($m^2m^{-2}$) | | |
|------|-----------|-----|------|------|------|------|-------|------|---------|---------|-----|
| | | min | Q25 | mean | Q75 | max | $A_b$ | $A_s$ | $LAI_s$ | $LAI_w$ | $LAI$ |
| N1 | 346 | < 1.3 | 2.1 | 3.8 | 4.4 | 65.4 | 25 | 19 | 7.5 | 2.3 | 5.2 |
| N2 | 196 | < 1.3 | 1.6 | 4.1 | 4.0 | 66.5 | 29 | 18 | 7.5 | 1.7 | 5.9 |
| S3 | 28 | 1.8 | 12.7 | 30.2 | 40.7 | 80.9 | 75 | 52 | 8.1 | 1.7 | 6.5 |
| S2 | 20 | 19.1 | 32.4 | 37.4 | 44.5 | 63.8 | 60 | 40 | 7.4 | 1.2 | 6.3 |
| S1 | 17 | 16.4 | 31.8 | 41.1 | 49.8 | 58.9 | 61 | 39 | 7.1 | 0.7 | 6.4 |

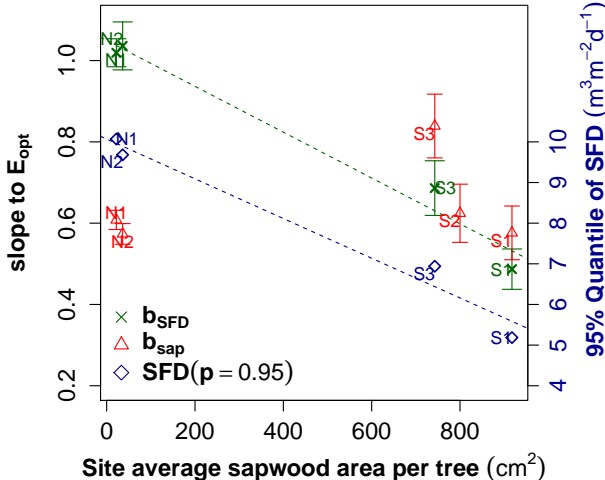

**Figure 7.** Sensitivity to $E_{opt}$ as a function of the site-average sapwood area per tree computed by $A_s/n_{tree}$. On the left y-axis: sensitivity of site-average sap flux density $b_{SFD}$ (green crosses, see also Table 3) and the sensitivity of the upscaled transpiration estimate $b_{sap}$ (red triangles). The vertical bars show the standard deviation of the sensitivity estimates tabulated in Table 2 and 3. Right y-axis: 95% quantile of daily $SFD$ maxima per site to estimate a robust maximum $SFD$. The dashed lines show linear regression model fits.

stand basal areas and thus total stand sapwood areas, which is the key factor in extrapolating $SFD$ to site level. Although most trees are found at the north-facing sites, their total sapwood area is about half of the south-facing sites. The valley site (S3) has the largest sapwood area (Table 5). The LAI measurements also show the largest values for the valley site and slightly lower values for the south-facing site. Lower values are found at the north-facing sites. However, the differences in LAI between sites are comparably small when compared with sapwood area.

Plotting $b_{SFD}$ as a function of the site-average sapwood area per tree ($A_s/n_{tree}$), see Figure 7, we find that $b_{SFD}$ decreases with $A_s/n_{tree}$. In addition, the maximum $SFD$ estimated as the 95% quantile of the daily maxima show a very similar decline.





Thus both measures show higher sap flux densities in the younger and smaller north-facing stands as compared to the older south-facing stands. In contrast the sensitivity of $E_{sap}$ is rather constant $b_{sap} \approx 0.6 \ [mm \ d^{-1}]/[mm \ d^{-1}]$ across sites at the hillslopes. Only at the valley site $b_{sap}$ is significantly higher with $\approx 0.8 \ [mm \ d^{-1}]/[mm \ d^{-1}]$. Although this site has a lower sapwood area per tree ($A_s/n_{tree}$) than the upper south-facing sites, the total sapwood area at S3 is larger, because of a few very tall trees, see Table 5.

## 4 Discussion

### 4.1 Using thermodynamic limits to estimate daily potential evaporation

Generally, there are two different physical limitations of atmospheric evaporative demand which have been used for modelling $E$, (i) by energy limitation and (ii) the mass exchange limitations which requires that the vapor is removed before it might condense again (Brutsaert, 1982). The approach of maximum convective power applied here combines both of these physical constraints and emphasizes the strong link between the energy balance and the strength of convective motion.

The maximum power-derived estimate of potential evaporation $E_{opt}$ is quite similar to the Priestley-Taylor formulation, but has two key differences. First, the Priestley-Taylor equation contains an additional empirical factor which has been interpreted as an effect of large scale advection or boundary layer dynamics (Brutsaert, 1982). The second difference is the use of net shortwave radiation in our model, as opposed to net radiation in the Priestley-Taylor equation, the latter of which is not an independent forcing, as it depends on the surface temperature. Using shortwave radiation has important practical advantages because $R_{sn}$ can be obtained from global radiation measurements and albedo information. Further topographic influences on incoming radiation can be directly computed by topographic radiation correction methods (Šúri and Hofierka, 2004). This also circumvents the necessity of collecting site-specific radiation data, which can be challenging above high vegetation like forests.

The derivation of $E_{opt}$ is based on a representation of an atmosphere which is in equilibrium with the underlying surface and the surface-atmosphere exchange of heat, moisture and momentum is driven by locally absorbed solar radiation (Kleidon and Renner, 2013b). This implies that meteorological variables such as wind speed or vapor pressure deficit, which are commonly used in Penman formulations, cannot change independently, but are rather constrained by land-atmosphere interaction. In order to test if these assumptions are generally met, we tested for the effect of VPD on transpiration in our dataset. The results showed that VPD and windspeed did not add consistently to the explained variance. In addition the correlation of the transpiration estimates to the FAO Penman-Monteith grass reference evaporation is on average slightly higher but follows the same patters across sites. Hence, the two additional input variables only slightly increased the predictability of atmospheric demand which is consistent with the residual regression analysis above. Both of these results indicate that $E_{opt}$ captures the dominant drivers of daily sap velocity variations within this temperate climate without requiring further data on wind speed and VPD.

The derivation of $E_{opt}$ shown in Section 2 was based on a range of simplifying assumptions to focus on the emerging maximum power limit in a coupled land-atmosphere system (Kleidon and Renner, 2013b; Kleidon et al., 2014). Most important



features, which need to be addressed in future work are (i) changes in heat storage, (ii) a revision of the simplified scheme for longwave radiative exchange, and (iii) horizontal circulation patterns by large scale and mesoscale circulation.

By assuming a steady state, heat storage effects were neglected in the derivation of the turbulent fluxes $J_{opt}$. However, heat storage in the surface-atmosphere system is an important mechanism to balance the seasonally and diurnally varying
input of solar radiation. A comparison of $J_{opt} = R_{sn}/2$ with the empirical net radiation estimate of Allen et al. (1998) which was used to estimate $E_{PM}$ showed that $J_{opt}$ is consistently lower during summer. Thus, to improve daily estimates based on thermodynamic limits, it is recommended to consider seasonal heat storage effects in the derivation. Heat storage is even more relevant at the diurnal time scale which reduces the applicability of $E_{opt}$ as given in Eqn 5. The relevance of considering heat storage is illustrated in Fig. 8 which shows a distinct hysteresis of both sap flow and air temperature with respect to global
radiation on a sunny day. Both sap flow and temperature linearly increase with shortwave radiation in the morning hours, but remain high after midday until sunset. Thus for the same amount of received global radiation there is a distinct difference between morning and afternoon transpiration. While moisture storage in soil and plants also affect the hysteresis, the dominant magnitude of the hysteresis is probably due to the lag of temperature to radiation leading to a VPD-radiation lag (Zhang et al., 2014). Hence for predictions at the diurnal time scale, heat storage effects as well as boundary layer dynamics have to be
accounted for.

Another limitation of the approach is the simple linearized scheme for longwave heat exchange. The description of radiative exchange affects the maximum power limit, because longwave radiation "competes" with the convective fluxes to cool the surface (Kleidon et al., 2015). Although the parameter describing the net longwave radiative exchange drops out in the maximization for the convective fluxes, the radiation scheme affects the partitioning of radiative and convective fluxes at the
surface. Therefore, a more detailed description of radiation transfer processes will increase the ability to predict surface energy partitioning.

The third limitation is concerned with the spatial representativity of $E_{opt}$, but also other potential evaporation formulations and the role of horizontal heat and mass exchange. While the land surface is generally quite heterogeneous, the lower atmosphere is mostly well mixed (Claussen, 1991). By using the topographically corrected incoming radiation we effectively treated
each site as an independent surface-atmosphere column. We thus neglected any lateral fluxes which could alter the estimated potential evaporation. The topographic correction increased the potential evaporation at the south- and decreased estimates for the north-facing sites. To evaluate this effect with respect to the transpiration estimates we used the global radiation measured at the nearby grassland site as forcing for all forest sites. The correlation to the transpiration estimates was unaffected, but the topographic correction slightly amplified the aspect related differences of the response to $E_{opt}$ between sites (Figure S1). For
a better understanding of such micro-climatic effects spatial observations of the canopy surface energy balance are required to test more detailed models which include horizontal exchange processes. Progress in this respect will be quite important for the parameterization of sub-grid processes in numerical land surface models.





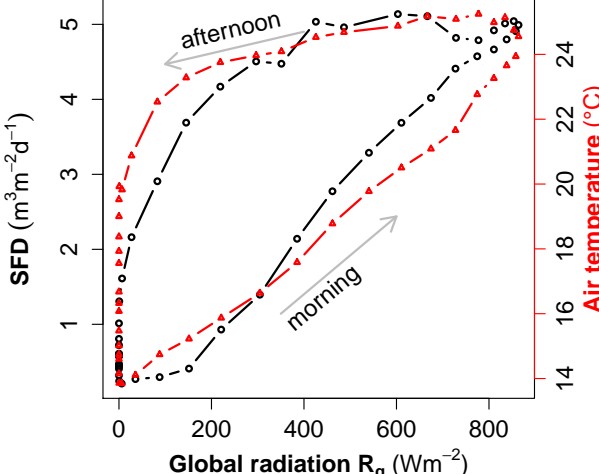

**Figure 8.** Hysteresis of sap flux density (left y-axis) and air temperature (right y-axis, red) plotted as a function of global radiation at the field site G1. Data are half-hourly values over one sunny summer day 2013-07-07. Sap flow data is taken from a beech (tree 1) at site S1.

## 4.2 Topographical and stand composition effects

Our results demonstrate a strong influence of daily variations in atmospheric demand on both transpiration estimates. This strong correlation at the daily timescale may allow us to separate the timescales of atmospheric demand and plant water limitations which may become relevant at timescales larger than one day. Differences in the slope of the relationship to $E_{opt}$

can then be compared across sites and be used to identify potential permanent site controls on transpiration.

Of the many factors that influence transpiration, stand composition and topography were the most important invariant controls during our measurement campaign. The measurement transect was placed on a valley cross-section to primarily reveal the influence of hillslope aspect. Our results indicate only significant effects of aspect and hillslope angle for the sensitivity of $SFD$ to $E_{opt}$, but no effects for the sensitivity of the upscaled transpiration estimate $E_{sap}$. $E_{RWU}$ shows a weak relation

to these topographic factors, which was, however, not significant because of the larger uncertainty of the soil moisture-based estimate.

The biometric measures of stand composition, however, also covary with aspect and hillslope inclination. Here we found a strongly negative linear relationship of $b_{SFD}$ and the maximum sap flux densities to the site-average sapwood area per tree (Figure 7). Such a decline in $b_{SFD}$ and maximum $SFD$ with sapwood area was found to be a univeral tree hydraulic

mechanism observed across different tree species (Andrade et al., 1998; Meinzer et al., 2001). Interestingly, Andrade et al. (1998) found that maximum sap velocity nonlinearly declined with sapwood area, with strongest effects at very small trees or branches with $A_S < 100 cm^2$. Considering such an increase in $b_{SFD}$ for the understorey vegetation at the north facings sites in our study would increase their relevance in estimating stand transpiration.

In addition to the tree size differences, our sites show marked differences in canopy structure. The open canopy structure

may explain the larger $b_{SFD}$ of the dominant trees sampled at the north-facing sites, because tree growth could be enhanced





by gaps in the canopy structure. For example Schweingruber et al. (2006) have shown that annual tree ring width of beech trees increase quickly after neighboring trees have been cut. At the south-facing sites average tree size is high but there is a low number of stems. Here individual tree growth is limited by the neighboring trees with similar demands for light, water, nutrients and space. Self-thinning or thinning by forest management has resulted in a more even-sized stand structure with

old and large trees. Thus, the closed canopy structure which is composed of well established trees may have lead to a more conservative water use per tree level which is reflected by low $b_{SFD}$ and low maximum sap flux densities. These arguments are consistent with results of studies that show higher sap flux densities of thinned forest stands as compared to control stands (Morikawa et al., 1986; Bréda et al., 1995; Nahm et al., 2006).

The upscaling procedure integrates both the tree distribution within the stand and the sap flux density observations to yield

the stand transpiration estimate $E_{sap}$. Apparently this has led to a compensation of the vastly different stand structure and sap flux densities across sites. Due to this tradeoff between $SFD$ and the sapwood area similar transpiration rates are achieved at the hillslope sites. Therefore, forest transpiration may indeed be regarded as a conservative hydrological process (Roberts, 1983) even in this very complex terrain. Only for the valley site S3 significantly larger $E_{sap}$ and $b_{sap}$ was estimated. This site has comparably high maximal $SFD$ and the largest total sapwood area, even though $A_s/n_{tree}$ is lower than observed at the

upper south-facing sites. The most likely reason for the comparably dense, but tall forest is the vicinity to the nearby stream which reduces potential soil water limitations. A similar upslope – downslope effect was recently also established by Kume et al. (2015) for a Mountainous stand of Japanese cypress with the lower site having taller trees with higher $SFD$ but same tree density.

### 4.3   Response to soil moisture

Due to low rainfall amounts, soil moisture decreased from July to September at all sites and all depths. Thus the soil moisture reduction may have limited transpiration. We captured this effect of soil moisture limitation of transpiration through the regression of the residuals of the transpiration to $E_{opt}$ relationships to site-average soil moisture conditions. Results showed that soil moisture explained a significant part of the residuals of $E_{sap}$ at the south-facing sites (Table 2), which implies that site-scale transpiration was also water limited during the dry period in August. At the north-facing sites water limitation at the site scale

was only apparent during a shorter period in September. However, we found significant residual correlation to soil moisture at three of four sampled small beech trees. This suggests that small trees are more susceptible to water shortage.

We argue that soil water limitation effects on transpiration could be topographically enhanced. Firstly, by aspect which affects the amount of received solar radiation and thus the atmospheric demand for water. A higher atmospheric demand increases evaporation from incepted water and from the soil which reduces the amount of precipitation entering the soil. Secondly, the

hillslope inclination could have enhanced lateral runoff at the steeper south-facing sites, which reduces the soil water holding capacity (Bronstert and Plate, 1997). These topographical factors of soil moisture availability are also apparent in the tree species composition of our sites. While the north-facing sites are predominantly composed of beech and a few spruce trees, the south-facing sites have about 10% oak trees, which are known to cope well at more dry sites (Zapater et al., 2011).





### 4.4 Limitations of transpiration estimates

#### 4.4.1 Limitations of sap flow observations

Generally, sap flow observations are not limited by spatial heterogeneity and complex terrain which would limit the applicability of micrometeorological measurements (Wilson et al., 2001). Installation is relatively simple and sensors are inexpensive.
Despite these advantages, we experienced the following limitations:

a) deriving water fluxes requires extrapolation from the point measurement at some specific place within the stem to the entire tree. However, sapwood conductivity can have radial and circumferential differences and species-specific properties (Wullschleger and King, 2000; Saveyn et al., 2008). This can easily bias sap flow estimates (Köstner et al., 1998; Shinohara et al., 2013; Vandegehuchte and Steppe, 2013). An indication of this problem is that we found different $b_{SFD}$ at the same tree installed the year before. Comparison with measurements taken at the same trees from the previous year showed differences in $b_{SFD}$ ranging between -0.39 and 0.23 with an average of 0.04 $[m^3 \, m^{-2} d^{-1}]/[mm \, d^{-1}]$ (estimated for 9 trees with more than 30 days of data each year).

b) there is a sample bias towards larger trees as the method is more difficult with very small trees which would require a different type of sensor, because the heat ratio method is designed only for lower sap velocities (Marshall, 1958; Burgess et al., 2001). This adds a sampling uncertainty in estimating site transpiration where much understorey exists. This is especially relevant for the north-facing sites, with a median DBH of 4 cm. This means that most trees were not sampled. The sampling rather reflected the trees which contributed most to the stand sapwood area.

c) The inter-comparison of sap flux density measured in different trees is limited by the fact that xylem characteristics in the estimation of $SFD$ are required (Burgess et al., 2001). Most important is the thermal diffusivity of sapwood, $k_w$ as used in Eqn. (7). This conductivity is a function of wood density and wood water content $m_c$ (Burgess et al., 2001; Vandegehuchte and Steppe, 2012a), both of which vary between species and trees (Gebauer et al., 2008). Xylem water content $m_c$, which in addition influences the apparent sap flux density through affecting the sapwood heat capacity (see Eq. 9), was shown to have seasonal changes in diffuse-porous species (Glavac et al., 1990; Hao et al., 2013). Glavac et al. (1990) found that $m_c$ can reduce to about 25% in sampled European beech trees during summer. Such a decline would thus reduce the apparent sap velocities. Therefore it is recommended to use methods that take into account wood moisture content changes (Vandegehuchte and Steppe, 2012b). Here, in the absence of measurements of such wood properties we used the same parameters for all trees.

#### 4.4.2 Limitations of root water uptake estimates from soil moisture variations

The advantage of using temporally highly resolved soil moisture readings is that it allows to estimate root water uptake without further information on soil properties (Guderle and Hildebrandt, 2015). The accuracy of the method depends on various factors that can influence results:

a) Data filtering: the method only applies under conditions with negligible soil water movement excluding events of infiltration, drainage, capillary rise or hydraulic redistribution. These fluxes can have major influences on the observations of soil moisture and comprise the second term of the right hand side of Eqn 14. Thus the estimates depend on the choice of suitable





filter criteria. A very strict filter would reduce the number of estimates, whereas soft filter criteria may result in biased $E_{RWU}$ estimates. Hence, seasonal or annual totals cannot be derived from this method alone. We use relatively strict filter criteria for night time fluxes of 0.1 Vol% which is close to the sensor resolution. This filter criteria set the maximum accuracy per soil layer of 200 mm depth to $0.2\ mm/d$.

b) Soil heterogeneities, dominant at the hillslopes, can induce large local variations in soil moisture and may lead to dissimilar / biased $E_{RWU}$ compared to other methods (Wilson et al., 2001). Here, we found that the influence of $E_{opt}$ and soil moisture content on $E_{RWU}$ varied between soil profiles at a specific site (Table S3 and Fig. S5). Large differences were observed at site N2 which results in a quite uncertain site-average estimate of $E_{RWU}$. Therefore it is recommended to install several, representative measurement profiles when such a soil water budget method is used for transpiration estimation in heterogeneous

terrain (Schwärzel et al., 2009).

c) Deep root water uptake in response to drying topsoil may cause root water uptake below the deepest measurement depth in forest sites as observed e.g. by Teskey and Sheriff (1996); Wilson et al. (2001). Observations from the deepest sensor profile confirm root uptake at 60 cm depth (Figure S4), which also persists during the dry period. However, overall the contribution of deep root water uptake is assumed to be small, given the low observed diurnal variations.

### 4.4.3   Upscaling sap flow to site scale transpiration

The estimates of site scale transpiration based on up-scaled sap flow measurements were of similar magnitudes and correlated well with estimates derived from soil moisture variations. The seasonal estimates by $E_{sap}$ are about $50\ mm$ lower than other estimates for beech forests. For example Schipka et al. (2005) found 200-300 mm per year for European beech forests in Germany. Their sites, however, have been located in less steep terrain.

The comparison between $E_{RWU}$ and $E_{sap}$ also revealed striking differences which could be an indication of the potential shortcomings of both methods, as discussed above. While the south-facing sites are in good agreement, $E_{sap}$ at the north-facing sites seems to be quite low. First, this is due to the low basal area at the north-facing site. One reason could be that the assumed sapwood area of the few tall trees is much larger than reported in the literature. Another possible reason could be that small trees ($< 8cm$ DBH), which were not sampled, had a significant contribution to stand transpiration. Also $E_{RWU}$ might

overestimate actual transpiration because soil evaporation would equally contribute to the diurnal signal in soil moisture. For example Bréda et al. (1993) also found consistently larger estimates of a soil water balance method than stand transpiration estimates, which was attributed to soil evaporation by the authors.

### 5   Conclusions

We aimed to infer the dominant temporal and spatial controls on forest transpiration along a steep valley cross-section through

ecohydrological measurements of sap flow and soil moisture and their relation to atmospheric evaporative demand. The estimation of transpiration in space and time for forests in complex terrain is a challenge in its own right. Obtaining transpiration is only possible through indirect observations, whereby each method has its own limitations. Therefore we used two independent





observations to obtain site-scale estimates of transpiration along the hillslope transect. To estimate atmospheric demand, a formulation similar to the well known Priestley-Taylor equilibrium evaporation concept was employed. The formulation is based on a simplified energy balance representation of the surface-atmospheric system and hypothesizing that convection operates at its upper thermodynamic limit. The formulation does not require empirical parameters and only requires data on the absorbed

solar radiation and temperature. We find that at the daily timescale this approach explains most of variability in both transpiration estimates at the site and tree scale. This suggests that atmospheric demand is the dominant control on daily transpiration rates in this temperate forest. Although the well-established FAO Penman-Monteith reference evaporation yields slightly higher correlation and 20-30% higher values, it requires additional data of net radiation, VPD and wind speed. Thereby both, VPD and wind speed did not add consistently to the explained variance and are also difficult to obtain above forests. While our

results demonstrate that thermodynamic limits provide a first-order estimate for potential evaporation, we have to stress that the derivation is based on the simplest possible energy balance representation. Further refinements will probably improve the predictability of surface exchange fluxes.

Despite the prevailing topographic contrasts between the north- and the south-facing measurement sites, we find that upscaled stand transpiration yields rather similar seasonal totals as well as a similar average response to atmospheric demand.

This similarity is achieved through a compensation of the low sapwood area with high sap flux densities at the north facing sites, while at the south-facing sites a high sapwood area was accompanied with low sap flux densities. It appears that individual and stand average sap flux densities can vary strongly in heterogeneous terrain in order to compensate for tree size and stand structural differences through tree hydraulic mechanisms. The importance of these stand structural differences on stand transpiration thus masks the potential effects of topographical factors such as aspect and hillslope angle which are cross

correlated. However, during dry periods we find that topographic factors can enhance the response of transpiration to soil water limitation.

We conclude that apart from the unavoidable limitations in estimating stand transpiration and potential evaporation in complex terrain, we find that relating the employed ecohydrological observations to a thermodynamically constrained estimate of atmospheric demand enables important insights in the temporal drivers of transpiration and how they vary at the hillslope scale.

*Acknowledgements.* This research contributes to the "Catchments As Organized Systems (CAOS)" research group (FOR 1598) funded by the German Science Foundation (DFG). We thank all people involved in the field work. In particular the technical backbone Britta Kattenstroth (GFZ Potsdam), Tatiana Feskova (UFZ - Leipzig), Laurent Pfister and François Iffly (LIST, Luxembourg) and the landowners for giving access to their land.



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
