# Peer review of "Dominant controls of transpiration along a hillslope transect inferred from ecohydrological measurements and thermodynamic limits"

_Hydrology and Earth System Sciences, 2015_

## Referee Comment (RC1) · Anonymous Referee #1 · 6 Feb 2016

**Review Comments on HESS-2015-535**

*"Dominant controls of transpiration along a hillslope transect inferred from ecohydrological measurements and thermodynamic limits"*

M. Renner, S. K. Hassler, T. Blume, M. Weiler, A. Hildebrandt, M. Guderle, S. J. Schymanski, and A. Kleidon

**Major Comments**:

This paper reports an important finding that transpiration is dominated by radiation forcing while VPD and wind speed adds no explanation power. The analysis based on field observations in the theoretical framework of maximum convective power principle is sound and convincing. This study opens new possibilities of modeling transpiration, which requires VPD and wind speed using the conventional methods. I recommend acceptance of the paper hoping that the finding is disseminated quickly in the Earth system modeling communities.

I suspect that the finding is more general than indicated by the paper title as the field data used in this study were collected "along a hillslope transection". Would the same conclusion be true over a flat land surface? Topography only affects the magnitude of surface net solar radiation as Eq (5) appears to hold regardless of topography. It is unclear the experiment sites were selected by design or convenience (existing observation facilities).

The word "atmospheric evaporation demand" is misleading. As shown in Eq (5), the atmospheric demand $E_{opt}$ is mainly due to net solar radiation that has little to do with the atmosphere.

**Editorial Comments**:

Page 8: "… diffusivity … $k_w = 2.5 \cdot 10^{-5}$ m s$^{-1}$ …" should be "… $k_w = 2.5 \times 10^{-5}$ m$^2$ s$^{-1}$ …".

Technical details of "Sap flow measurements", "Upscaling of sap flow to tree and stand transpiration" and "Root water uptake estimation" may be put in appendices.

Text narrative can be substantially shortened. For example, large part of the conclusion may be either cut down or put in "Discussion", which is better shortened as well.

---

## Author Comment (AC1) · 12 Feb 2016

We thank the reviewer for his positive and encouraging comment on the manuscript. Although the Reviewer strongly supports the one of our mains findings that "transpiration is dominated by radiation forcing while VPD and wind speed adds no explanation power.", we want to emphasize that these findings are valid for this temperate forest. Under certain conditions vapor pressure deficit (VPD) and wind speed may contribute to the explained variance. For example very moist conditions with very low VPD could reduce transpiration. Also very dry conditions with high VPD could enhance potential transpiration rates. Such conditions were rarely found at our sites and therefore did not add to the explained variance. But clearly, these aspects should be investigated further

by a larger and more diverse data set of different climatic and topographic conditions.

The research focus of this manuscript was to determine the dominant controls on transpiration. Apart from the strong control of radiation, we found that transpiration rates and their response to potential evaporation were rather similar on the upslope sites, despite different aspect and stand structure. Only the downslope site with access to riparian water reached larger transpiration rates. The research design of the observation sites was structured to enable such comparisons to reveal the effects of topography on hydrological functioning (Zehe et al., 2015, HESS).

During the revision process we will take the advise of the referee to shorten the manuscript within the context of the comments of all reviews.

References Zehe, E., U. Ehret, L. Pfister, T. Blume, B. Schröder, M. Westhoff, C. Jackisch, et al. 2014. "HESS Opinions: From Response Units to Functional Units: A Thermodynamic Reinterpretation of the HRU Concept to Link Spatial Organization and Functioning of Intermediate Scale Catchments." Hydrol. Earth Syst. Sci. 18 (11): 4635–55. doi:10.5194/hess-18-4635-2014.

---

## Referee Comment (RC2) · Anonymous Referee #2 · 28 Mar 2016

**Reviewer(s)' Comments to Author:**

The manuscript entitled "Dominant controls of transpiration along a hillslope transect inferred from ecohydrological measurements and thermodynamic limits" consist of an experimental study that investigates what controls summer transpiration rates in a deciduous forest located along a set of mild hillslope transects in a European temperate climate (energy-limited during the winter season). In addition to a set of measured ecohydrological variables (namely sapflow and soil moisture), transpiration estimates were compared versus potential evaporation using a parsimonious model based on thermodynamic principles.

Overall, I really liked the set-up of this study: very clear and structured. The methodology is also sound. Therefore, I have no big comments on the general approach, although several smaller remarks remain. Some of the references are a bit old and some recent publications were suggested. Despite this, I recommend this manuscript to be accepted pending a minor revision. I am confident the authors can address these issues stated in the following review.

**ABSTRACT**

The abstract would improve if you make it in 3 paragraphs (introduction, hypothesis/model, results/take home messages). The way you write the latter paragraph is very important. If I understood correctly, the dominant temporal controls of transpiration (Et) are Ta and Rn (hypothesis that is supported by your results) whereas spatial controls mainly consist of soil moisture availability (was this a priori hypothesis too? Are you sure there are no previous studies that investigate the spatial controls of Et?). If so, then you can say that diel patterns in Et are primarily governed by local climatic conditions but spatial heterogeneities may arise from landscape factors (tree density, topography gradients). Or, from a modeling perspective you could say that Ta and Rn are sufficient for point/plot

scale Et estimation but as your spatial scale goes up → hillslope/catchment, landscape and vegetation factors become important for Et assessment.

Line 9: Finding from recent studies (See Tim McVicar and Mike Roderick seminal papers on multi-decadal pan-evaporation decline) suggest that wind speed is a key parameter controlling long term AED so I would make a brief mention in the abstract (perhaps saying that although the thermodynamic approach works fine using T and Rn, other parameters such as wind speed are important, especially for long-term scales) and in the manuscript too (adding one paragraph should be enough).

1. Introduction

P2L4: Recent evidence has shown that ecosystems can be sustained by water stored in rock layers (see Schwinning, 2010; Oshun et al., 2016) so it would be worth to mention that Et can no longer be solely attributed to water stored in the soil matrix. Also, do not forget that groundwater can be the foremost Et source in arid catchments (see Miller et al., 2010)

P2L5 and L6: Please cite one or two papers about the effects of Et on the energy and water balance. What is the range of Et losses on the water balance? Can you provide a range of estimates?

P2L8: Since your work explores the role of Ta and Rn on Et, perhaps you could add that climatic processes (in addition to biogeophysical controls) also influence the temporal variation of Et.

P2L10: Something is missing in your first sentence. Your manuscript is intended to be published in a hydrology journal so say something related to it. Maybe that it is important to investigate the dominant controls of Et in complex terrain as pretty

much of Europe's forests are located over complex terrain areas. Also, add a reference.

P2L11: "that the first order controls". Do you mean controls of land use change?

P2L12: "Hillslope angle" -> "hillslope" should be written in small caps.

P2L13: You references date back from the 60´s! It would be nice to include something more recent and within an ecohydrological perspective. Perhaps Ivanov et al., 2009?

P2L13: "alter lateral distribution of water" → alter the lateral distribution of water

P2L16: "of supply and demand" → of (water and energy) supply and demand.

P2L16: In addition to Holst et al. work, I suggest you read and cite Link et al. paper.

P2L30: "Contrary to the classic notion of Dalton evaporation" → Contrary to the classic formulation of Dalton (add citation).

P3L1: "applied for long-term annual means" → applied for long-term annual estimates of E.

P3L9: "how the response to atmospheric demand changes" → how much they respond to atmospheric demand changes.

P3: Somewhere in the last paragraph of your introduction, you should clearly state the aims/objectives of your manuscript.

**2. Methods**

**P3L30 Question: If you have meteorological observations in each site, why are you using only one dataset for model forcing? I assume it is because the datasets in each site look quite similar, if so, you should say that no significant differences were found between the 6 datasets.**

**P4L2: Your aim should be moved earlier at the end of the introduction section. Also, what do you mean by "potential evaporation of a surface at saturation from first principles"? Perhaps using the term "potential evaporation" is more clear to your readership.**

**P4L5: "grid-scale global predictions" → grid-based global scale predictions**

**P4L13: There is an issue after the description of the heat flux partitioning. It reads as if the latent heat flux is equal to the sum of the heat fluxes.**

**P5L4: "namely the temperature gradient $T_s − T_a$ responds"→ namely the temperature gradient which responds.**

**P5L11: "this limits (subscript opt)" → this limit.**

**P8L3: "35 and 42 yrs" → yrs old.**

**P8L5: "suspect data was filtered" -> data were filtered.**

**P8L9: "A arithmetic mean" → An arithmetic.**

**P8L12:** "The circumference at breast height of all trees with circumference larger or equal 4 cm" → with a circumference larger or equal to 4 cm.

**P8L15:** "LICOR LAI-2200" → LICOR LAI-2200 Plant Canopy Analyzer

**P8L23:** "Following the manufacturer manual we assigned for each sensor depth an representative" → Following the user's manual we assigned for each sensor depth a representative.

**P9L7:** The acronym of diameter at breast height (DBH) should be defined here, and not until line 9.

**P10L22:** It was not clear at which effective depth was the root water uptake considered. Do you have independent observations of the depth reached by tree roots?

---

## Editor Comment (EC1) · T.A. Bogaard (Editor) · 30 Mar 2016

Dear authors

you received two positive reviews with minor revisions required. I am pleased to let you know the manuscript can be published taking into account these minor revisions.

I look forward to your reply to the referees remarks, suggestions, questions and revised manuscript in due time.

Kind regards, Thom

---

## Author Response (AR1)

**Response to the reviewer comments**

Maik Renner

April 14, 2016

**1  Comments and reply to reviewer #1**

**This paper reports an important finding that transpiration is dominated by radiation forcing while VPD and wind speed adds no explanation power. The analysis based on field observations in the theoretical framework of maximum convective power principle is sound and convincing. This study opens new possibilities of modeling transpiration, which requires VPD and wind speed using the conventional methods. I recommend acceptance of the paper hoping that the finding is disseminated quickly in the Earth system modeling communities. I suspect that the finding is more general than indicated by the paper title as the field data used in this study were collected "along a hillslope transection". Would the same conclusion be true over a flat land surface? Topography only affects the magnitude of surface net solar radiation as Eq (5) appears to hold regardless of topography. It is unclear the experiment sites were selected by design or convenience (existing observation facilities).**

We thank the reviewer for his positive comments which helped to formulate the take home message more clearly. We already replied to the comments in a separate author reply `http://doi:10.5194/hess-2015-535-AC1`.

- **The word "atmospheric evaporation demand" is misleading. As shown in Eq (5), the atmospheric demand $E_{opt}$ is mainly due to net solar radiation that has little to do with the atmosphere.**

  In using the term "atmospheric evaporation demand", we follow common terminology in the literature, where various subsets of atmospheric drivers have been considered in its definition (Federer, 1982; Calder et al., 1983; McVicar et al., 2012; McMahon et al., 2013). In our paper, $E_{opt}$ is derived from a surface-atmosphere system, where we treat the atmosphere as a heat engine and maximize the power of convective motion. This maximum in power leads to a form where atmospheric evaporative demand only depends on absorbed solar radiation and surface temperature. Nevertheless the process of atmospheric convection interacting with the surface is the key of our approach.

- **Page 8: "... diffusivity ... $k_w = 2.5 \cdot 10^{-5} m\, s^{-1}$ ..." should be "... $k_w = 2.5 \cdot 10^{-5} m^2\, s^{-1}$ ...".** Done.

- **Technical details of "Sap flow measurements", "Upscaling of sap flow to tree and stand transpiration" and "Root water uptake estimation" may be put in appendices.** Thank you for the suggestion, but since the information given here is relevant for understanding the results, we believe it is appropriate to be included in the main text, however, we leave the final decision to the Editor.

- **Text narrative can be substantially shortened. For example, large part of the conclusion may be either cut down or put in "Discussion", which is better shortened as well.** We agree that the text narrative could be shortened, but given the complexity of interacting processes discussed in this paper, we chose not to remove redundancy but repeat the main points in the conclusions. To clarify that this section actually contains a summary of the main findings, we renamed it to "Summary and conclusions"

**2 Comments and reply to reviewer #2**

**The manuscript entitled "Dominant controls of transpiration along a hillslope transect inferred from ecohydrological measurements and thermodynamic limits" consist of an experimental study that investigates what controls summer transpiration rates in a deciduous forest located along a set of mild hillslope transects in a European temperate climate (energy-limited during the winter season). In addition to a set of measured ecohydrological variables (namely sapflow and soil moisture), transpiration estimates were compared versus potential evaporation using a parsimonious model based on thermodynamic principles. Overall, I really liked the set-up of this study: very clear and structured. The methodology is also sound. Therefore, I have no big comments on the general approach, although several smaller remarks remain. Some of the references are a bit old and some recent publications were suggested. Despite this, I recommend this manuscript to be accepted pending a minor revision. I am confident the authors can address these issues stated in the following review.**

We appreciate the encouraging and constructive comments of reviewer #2 which helped to improve the readability of the manuscript. Below we provide a detailed response to the remarks of the reviewer.

- **The abstract would improve if you make it in 3 paragraphs (introduction, hypothesis/model, results/take home messages). The way you write the latter paragraph is very important. If I understood correctly, the dominant temporal controls of transpiration (Et) are Ta and Rn (hypothesis that is supported by your results) whereas spatial controls mainly consist of soil moisture availability (was this a priori hypothesis too? Are you sure there are no previous studies that investigate the**

**spatial controls of Et?). If so, then you can say that diel patterns in Et are primarily governed by local climatic conditions but spatial heterogeneities may arise from landscape factors (tree density, topography gradients). Or, from a modeling perspective you could say that Ta and Rn are sufficient for point/plot scale Et estimation but as your spatial scale goes up hillslope/catchment, landscape and vegetation factors become important for Et assessment.**

We thank the reviewer for his useful suggestion and added "take home" messages at the end of the abstract and the conclusions. We added the following paragraph at the end of the abstract:

"We conclude that absorption of solar radiation at the surface forms a dominant control for turbulent heat and mass exchange and that vegetation across the hillslope adjusts to this constraint at the tree and stand level. These findings should help to improve the description of land surface-atmosphere exchange at regional scales."

- **Line 9: Finding from recent studies (See Tim McVicar and Mike Roderick seminal papers on multi-decadal pan-evaporation decline) suggest that wind speed is a key parameter controlling long term AED so I would make a brief mention in the abstract (perhaps saying that although the thermodynamic approach works fine using T and Rn, other parameters such as wind speed are important, especially for long-term scales) and in the manuscript too (adding one paragraph should be enough).**

Roderick and Farquhar (2002) show that declining trends in pan evaporation can be largely explained by decreases in surface shortwave radiation using an equilibrium evaporation formula. This is thus inline with our findings. Clearly a decrease in solar radiation may also change land-atmosphere exchange and thus wind speed.

To clarify our arguments, also in response to the main comment of the first reviewer, we added a paragraph to the discussion in section 4.1 at P20L30.

"The dominance of absorbed solar radiation in explaining latent and sensible heat fluxes was also found by (Best et al., 2015), who showed that simple linear regression models with solar radiation and temperature as input and thus without any information on wetness conditions had a similar performance than commonly used land-surface models at 20 diverse tower flux sites. However, an improved reproduction of sensible and latent heat fluxes was obtained by Best et al. (2015) when relative humidity was included into their regression model. It is likely that our potential evaporation estimate $E_{opt}$ represents evaporation in a wet environment, whereas in drier environments, air humidity may become increasingly important. Interestingly, in contrast to our approach of eliminating air humidity while considering air temperature as an independent variable, Aminzadeh et al. (2016) estimated wet environment evaporation potential as a function of

radiation and air humidity, while eliminating air temperature. Clearly, more investigations, encompassing larger and more diverse data sets, are needed to better understand general patterns of atmospheric control on transpiration. For example, it may well be that air humidity carries information about soil moisture and hence adds to explanatory power of transpiration models in water limited environments, while radiation represents the main control in energy-limited environments. In how far wind, air humidity and temperature are affected by the land-atmosphere coupling and can hence be treated as internal variables, likely also depends on the scale of interest. "

- **P2L4: Recent evidence has shown that ecosystems can be sustained by water stored in rock layers (see Schwinning, 2010; Oshun et al., 2016) so it would be worth to mention that Et can no longer be solely attributed to water stored in the soil matrix. Also, do not forget that groundwater can be the foremost Et source in arid catchments (see Miller et al., 2010)** We included the suggestions as follows: "... soil, groundwater and possibly bedrock (Shuttleworth, 1993; Miller et al., 2010; Schwinning, 2010)."

- **P2L5 and L6: Please cite one or two papers about the effects of Et on the energy and water balance. What is the range of Et losses on the water balance? Can you provide a range of estimates?** We added references (Oke, 1987; Federer, 1973; Jasechko et al., 2013) but do not report quantitative estimates on the amount of $E_T$ because these might vary wrt. climate and ecosystem.

- **P2L8: Since your work explores the role of Ta and Rn on Et, perhaps you could add that climatic processes (in addition to biogeophysical controls) also influence the temporal variation of Et.** Done.

- **P2L10: Something is missing in your first sentence. Your manuscript is intended to be published in a hydrology journal so say something related to it. Maybe that it is important to investigate the dominant controls of Et in complex terrain as pretty much of Europe's forests are located over complex terrain areas. Also, add a reference.** We revised the first sentence to generally state that forests are often found in complex terrain. Unfortunately we did not found an assessment in the literature of how much of forested area is actually found in complex terrain in Western Europe. The importance of considering topographic controls on $E_T$ is then highlighted in the following part of the paragraph.

- **P2L11: "that the first order controls". Do you mean controls of land use change?** No, first order physical controls (water and energy). We revised this accordingly.

- **P2L12: "Hillslope angle" -> "hillslope" should be written in small caps.** done

- **P2L13: Your references date back from the 60s. It would be nice to include something more recent and within an ecohydrological perspective. Perhaps Ivanov et al., 2009?** We thank the reviewer for the good literature suggestion of Ivanov et al., 2009.

- **P2L13: "alter lateral distribution of water" alter the lateral distribution of water** done

- **P2L16: "of supply and demand" -> of (water and energy) supply and demand.** We revised this accordingly.

- **P2L16: In addition to Holst et al. work, I suggest you read and cite Link et al. paper.** We thank the reviewer for this interesting reference. The focus of the mentioned paragraph of the manuscript is to state that there is still a large uncertainty in how we model forest transpiration in sloped terrain, as illustrated by Holst et al. (2010). However, the mentioned paper by Link et al. (2014) focuses on the seasonal course of tree transpiration and how it differs among species, which is not the focus of our study.

- **P2L30: "Contrary to the classic notion of Dalton evaporation" → Contrary to the classic formulation of Dalton (add citation).** Done.

- **P3L1: "applied for long-term annual means" → applied for long-term annual estimates of E.** Done

- **P3L9: "how the response to atmospheric demand changes" → how much they respond to atmospheric demand changes.** Done

- **P3: Somewhere in the last paragraph of your introduction, you should clearly state the aims/objectives of your manuscript.** We now state our objective in the beginning of the third paragraph of the introduction.

- **P3L30 Question: If you have meteorological observations in each site, why are you using only one dataset for model forcing? I assume it is because the datasets in each site look quite similar, if so, you should say that no significant differences were found between the 6 datasets.** We only used the radiation and temperature data from the open field site (G1). Although radiation and temperature are measured as well in the forest sites, the instruments are mounted below the forest canopy. This makes radiation data for model forcing useless. Also air temperature below the canopy is impacted by being below the canopy. Therefore we chose to use data from the open field site, with radiation corrected by topography, for estimating $E_{opt}$.

- **P4L2: Your aim should be moved earlier at the end of the introduction section. Also, what do you mean by "potential evaporation of a surface at saturation from first principles"? Perhaps using the term "potential evaporation" is more clear to your readership.** We adopted the suggestion and just use the term "potential evaporation".

- **P4L5: "grid-scale global predictions"** → **grid-based global scale predictions** Done

- **P4L13: There is an issue after the description of the heat flux partitioning. It reads as if the latent heat flux is equal to the sum of the heat fluxes.** We revised this sentence.

- **P5L4: "namely the temperature gradient Ts-Ta responds" namely the temperature gradient which responds.** Revised accordingly.

- **P5L11: "this limits (subscript opt)" this limit.** Done.

- **P8L3: "35 and 42 yrs" yrs old.** Done

- **P8L5: "suspect data was filtered" -> data were filtered.** Done

- **P8L9: "A arithmetic mean" An arithmetic.** Done

- **P8L12: "The circumference at breast height of all trees with circumference larger or equal 4 cm"** → **with a circumference larger or equal to 4 cm.** Done

- **P8L15: "LICOR LAI-2200"** → **LICOR LAI-2200 Plant Canopy Analyzer** Done

- **P8L23: "Following the manufacturer manual we assigned for each sensor depth an representative" Following the user's manual we assigned for each sensor depth a representative.** Done

- **P9L7: The acronym of diameter at breast height (DBH) should be defined here, and not until line 9.** Done

- **P10L22: It was not clear at which effective depth was the root water uptake considered. Do you have independent observations of the depth reached by tree roots?** We revised P10L16: "The sum of $E_{RWU,d,z}$ obtained for each soil layer with a sensor gives a total daily root water uptake $E_{RWU,d}$ per profile." → "We then added $E_{RWU,d,z}$ of each soil layer to obtain a daily root water uptake $E_{RWU,d}$ per profile. For most profiles this allowed to estimate $E_{RWU,d}$ up to a depth of 600 mm, and for three profiles up to 800 mm (see Table S1 for an overview of sensor placement). During sensor installation thick roots were mostly found until a depth of $\approx 300\ mm$ with sporadic fine roots up to 800 mm depth."

[8]removed: notion of Dalton evaporation

[9]removed: are rather dependent variables which

[10]removed: means

[11]removed: how the response

[12]removed: changes

**2 Methods**

**2.1 Site description**

We analyze measurements at 6 different sites along a well instrumented steep forested hillslope transect (north- vs. south-facing, see Fig. 1) in the Attert catchment in Luxembourg over the vegetation period of 2013. The hillslope transect is part of the CAOS field observatory (Zehe et al., 2014) and is located in the western part of Luxembourg (5E°48'13", 49N°49'34") at about 460 m NN. The land cover of the transect is a mixed forest dominated by European beech (Fagus sylvatica L.). The north-facing slope has an inclination of $\approx 15°$ and is composed of a few dominant trees with large gaps and dense understorey mainly of young beech trees, whereas the south-facing slope is generally steeper ($\approx 22°$) and has no understorey and a denser canopy. Also tree species composition varies between slopes, with 97% beech on the north-facing sites with single spruce trees and 90% beech and 10% oak on the south-facing sites. The valley site has 80% beech with 10% spruce and alder, respectively. Geologically, the site is situated in northeast-southwest-trending fold system of Schists of the Ardennes Massif. The shallow soils developed on periglacial slope deposits (Juilleret et al., 2011). The deposits are generally found at a depth of 70-90 cm.

Standard meteorological data, global radiation, air temperature, relative humidity and wind speed were measured 2 m above ground at all sites. For the meteorological forcing we used the data from the open grassland site G1 which is located 240 m to the northwest of the forest site N1, see Fig. 1. Absorbed solar radiation $R_{sn} = (1 - \alpha)R_g$ in $Wm^{-2}$ is derived from global radiation $R_g$ and an albedo estimate of $\alpha = 0.15$ which is representative for deciduous forests (Oke, 1987).

**2.2 Estimation of atmospheric demand**

Our aim is to estimate the potential evaporation [..[13] ]from first principles and with few, independent input data. Therefore we make use of the concept of thermodynamic limits of convection which was recently established by Kleidon and Renner (2013b) and used successfully to estimate the sensitivity of the hydrologic cycle to global warming (Kleidon and Renner, 2013a; Kleidon et al., 2015) and for [..[14] ]grid-based global scale predictions of annual average terrestrial evaporation (Kleidon et al., 2014). Here we only illustrate how the concept is used to estimate potential evaporation, for further details the reader is referred to the mentioned publications.

Convection can be thought of as a heat engine which converts a temperature gradient into kinetic energy (Ozawa et al., 2003). To capture the fundamental trade-off of thermodynamic limits of convective exchange, we consider a simple two-box surface-atmosphere system in steady state, which is sketched in Fig. 2. We consider the steady state energy balance of the surface $R_{sn} = J + R_{ln}$. The surface is heated by absorption of incoming solar radiation $R_{sn}$. The turbulent heat fluxes $J$ and the net longwave exchange $R_{ln}$ both cool the surface. The turbulent heat fluxes are [..[15] ]composed of the sensible ($H$) and latent heat flux $\lambda E$[..[16] ]. Longwave radiative exchange is represented by a simplified linearized radiation $R_{ln} = k_r(T_s - T_a)$, with $T_s$ being the temperature of the surface and $T_a$ the temperature of the atmosphere and $k_r$ being a constant radiative
* * *
[13]removed: of a surface at saturation

[14]removed: grid-scale global

[15]removed: comprised by

[16]removed: : $J = H + \lambda E$

[revised manuscript text omitted]

---

## Author Response (AR2)

**Revision notes of the corrections for the final version**

Maik Renner

May 4, 2016

The editor and the first reviewer asked for reduction of the length of the methods section. We accomplished this by moving the technical details of sections 2.4,2.6, and 2.7 to the appendix. We kept the important parts in the methods section and combined the sections dealing with sapflow and upscaling. This step indeed shorted the main manuscript by about 3 pages and still provides the most important information for the reader.

A track changes version is attached.

[revised manuscript text omitted]
 | 66 | 130 | 1.30 ± 0.05 *** | 0.09 ± 0.13 | 0.88 | 0.88 | 0.03 | 0.07 ** | 0.11 ** |
| N1 | 2 | beech | 58 | 130 | 1.26 ± 0.04 *** | -0.06 ± 0.08 | 0.92 | 0.84 | 0.00 | 0.03 * | -0.01 |
| N1 | 3 | beech | 9 | 130 | 0.88 ± 0.05 *** | -0.08 ± 0.09 | 0.82 | 0.89 | 0.03 | -0.00 | 0.08 * |
| N1 | 4 | beech | 8 | 130 | 0.63 ± 0.08 *** | -0.22 ± 0.08 ** | 0.79 | 0.88 | -0.00 | 0.02 | 0.29 *** |
| N1 | | | | 130 | 1.02 ± 0.03 *** | -0.07 ± 0.07 | 0.90 | 0.91 | 0.01 | 0.01 | 0.00 |
| N2 | 1 | beech | 53 | 97 | 1.31 ± 0.08 *** | 0.23 ± 0.16 | 0.83 | 0.83 | -0.01 | -0.00 | 0.03 |
| N2 | 2 | beech | 49 | 97 | 1.64 ± 0.09 *** | -0.06 ± 0.18 | 0.87 | 0.88 | 0.00 | -0.00 | -0.01 |
| N2 | 3 | beech | 10 | 97 | 0.68 ± 0.04 *** | -0.11 ± 0.05 * | 0.89 | 0.82 | 0.01 | 0.00 | -0.00 |
| N2 | 4 | beech | 8 | 97 | 0.51 ± 0.08 *** | -0.14 ± 0.08 | 0.80 | 0.78 | -0.01 | 0.00 | 0.26 ** |
| N2 | | | | 97 | 1.04 ± 0.06 *** | -0.02 ± 0.11 | 0.87 | 0.88 | -0.00 | -0.01 | 0.02 |
| S1 | 1 | beech | 43 | 130 | 0.81 ± 0.08 *** | -0.23 ± 0.10 * | 0.77 | 0.80 | -0.01 | 0.01 | 0.34 ** |
| S1 | 2 | oak | 40 | 130 | 0.26 ± 0.01 *** | -0.10 ± 0.02 *** | 0.92 | 0.94 | 0.05 * | 0.01 | -0.01 |
| S1 | 3 | beech | 39 | 97 | 0.39 ± 0.12 ** | -0.14 ± 0.12 | 0.64 | 0.81 | 0.01 | 0.01 | 0.40 *** |
| S1 | 4 | beech | 46 | 130 | 0.40 ± 0.05 *** | -0.13 ± 0.06 * | 0.76 | 0.87 | -0.01 | 0.01 | 0.40 *** |
| S1 | | | | 130 | 0.49 ± 0.05 *** | -0.17 ± 0.06 ** | 0.80 | 0.84 | -0.01 | 0.00 | 0.30 ** |
| S3 | 1 | beech | 45 | 124 | 0.52 ± 0.06 *** | -0.22 ± 0.07 *** | 0.83 | 0.86 | -0.01 | 0.02 | 0.22 *** |
| S3 | 2 | beech | 39 | 124 | 0.86 ± 0.08 *** | -0.35 ± 0.10 *** | 0.84 | 0.90 | -0.00 | -0.00 | 0.09 * |
| S3 | | | | 124 | 0.69 ± 0.07 *** | -0.28 ± 0.08 *** | 0.84 | 0.87 | -0.01 | 0.01 | 0.12 * |

[revised manuscript text omitted]